# Characterization of indigenous chicken phenotypes in Liban Jawi District, Ethiopia: *A qualitative and quantitative analysis*

**Desalegn Begna**[1]*, **Teferi Bacha**[2], **Shamble Boki**[2], **Kasahun Bekana**[2]

**1** Policy Institute Study Institute, Agriculture and Rural Development Center, Addis Ababa, Ethiopia,
**2** Department of Animal Science, College of Agriculture and Veterinary Science Ambo University Mamo Mezemir Campus, Guder, Ethiopia

☯ These authors contributed equally to this work.
* dbegna67@gmail.com

**Data Availability Statement:** All relevant data are within the manuscript and its Supporting Information files uploaded in the additional information.

## Abstract

Indigenous chickens play a crucial role in the livelihoods of smallholder farmers in rural Ethiopia. This study aimed to phenotypically characterize indigenous chickens in the Liban Jawi district, focusing on measurements of phenotypic characteristics. The multi-stage sampling method selected 192 households with at least two mature indigenous chickens from 2,166 households, resulting in the sampling of 224 chickens (138 females and 86 males) for phenotypic characterization. Qualitative trait analysis revealed that male chickens exhibited a plain head shape (54.7%), single combs (39.5%), red feather plumage (26.7%), yellow shanks (54.7%), red earlobes (51.6%), and white skin (68.6%). Female chickens showed single combs (75.4%), red earlobes (51.4%), yellow shanks (50.1%), and brown mottled feathers (23.9%), with red (23.1%) and black (9.4%) feathers observed. Quantitatively, cocks had an average body weight of 1.46±0.02 kg, while hens weighed 1.21±0.01 kg. Cocks exhibited larger body dimensions, with significant measurements in body length (38.05±0.26 cm) and wingspan (38.56±0.13 cm). For hens, the average body length was 31.55±0.33 cm. Shank length emerged as a moderate predictor of body weight for both sexes (r = 0.45 for hens and r = 0.44 for cocks), indicating it should be combined with other factors for accurate assessments. Cluster and multivariate analysis revealed distinct phenotypic groupings among the indigenous chickens, highlighting significant variations in both qualitative and quantitative traits. This suggests the potential for improvement through selective breeding at the community level, influenced by unique environmental conditions and practices. These findings provide valuable insights into the phenotypic characterization of indigenous chickens, serving as a foundation for future breeding programs and conservation efforts aimed at enhancing productivity and preserving genetic diversity. Further molecular-level characterization is recommended to validate the current phenotypic results.

**Funding:** The author(s) received no specific funding for this work.

**Competing interests:** The authors have declared that no competing interests exist.

## Introduction

Indigenous chicken (*Gallus domesticus*) constitutes approximately 80% of the global poultry population, with the majority found in developing countries [1]. In Africa, village fowl, including indigenous chickens, make up more than 85% of the total fowl population. Ethiopia, in particular, has a significant chicken population, estimated at around 56.99 million, with approximately 78.85% being indigenous breeds [2]. The high proportion of indigenous chickens highlights their importance as a genetic resource in the country. Within Ethiopia, the West Shewa zone, including the Liban Jawi district, contributes about 1.3 million chickens to the total population of 16.7 million in Oromia [2]. Indigenous chickens play a crucial role in the livelihoods of smallholder farmers in rural Ethiopia [3,4]. These chickens are well-adapted to the local environmental conditions and serve as an important source of income, protein, and cultural significance for resource-poor households [5,6]. However, the productivity and genetic diversity of indigenous chickens in Ethiopia are threatened by various factors, including uncontrolled crossbreeding, disease outbreaks, and the introduction of commercial hybrid chickens [7,8].

According to [9], each village household typically keeps between 5 and 20 indigenous chickens, and [10] reported 7 to 10 birds per household,13.68±0.62 for Mekela areas [11], 13.59 mean for Gambella region [10]. These native chickens generally exhibit poor egg production performance, delayed maturity, and extended broodiness, as they have primarily been selected for their adaptive traits [12]. Indigenous chicken populations in Ethiopia have been characterized based on various criteria, such as plumage color and geographic region of sampling [7,13,14]. Understanding the unique traits, adaptation to the environment, and socio-cultural importance of indigenous chickens is essential for their conservation and utilization [8,7,15]. Phenotypic variations in major morphological traits, including plumage color, feather contours, shank and ear-lobe colors, and comb types, are common among indigenous chicken populations [7,10,16–18].

Characterizing and classifying indigenous non-descriptive chicken ecotypes is crucial for developing sustainable conservation and improvement strategies [15,17,19,20]. Although previous studies have documented the phenotypic diversity of indigenous chickens across various regions of Ethiopia, including well-known ecotypes such as Horro and Koekoek, much of this research has been limited and does not fully explore the genetic resources available at the local level [10,13,16,17,20–23]. Specifically, there is a need for more detailed investigations into the unique phenotypic traits of indigenous chickens in regions like the Liben Jawi district, which likely represent the Horro and Koekoek populations [24].

However, there has been limited research on the characterization and classification of indigenous non-descriptive chicken ecotypes, and efforts to identify and evaluate these genetic resources are still in their early stages [18].

Therefore, this study aims to phenotypically characterize indigenous chickens in the Liben Jawi district, focusing on both qualitative and quantitative traits, to document their unique physical characteristics and identify desirable traits for genetic improvement, thus laying the groundwork for effective conservation and sustainable utilization of these valuable local poultry genetic resources.

## Methodologies

### The study areas

The study was conducted in the Liban Jawi district in the West Shewa zone of the Oromia Regional State, Ethiopia. According to the district's Agricultural and Cooperatives Office Report, Liban Jawi district is geographically located between 8˚50'58" to 8˚54'4"Latitude/

North and 37˚22'21" to 37˚37'56"Longitude/East [25]. The district was purposively selected based on its potential for the indigenous chicken population and the rapid distribution of crossbred and exotic chicken breeds, which could impact indigenous chickens' population size and adaptive traits. The district is approximately 171 km west of Addis Ababa and 51 km from the zone administration Ambo town. It shares borders with the Mida Kegni, Jibat, Toke Kutaye, and Chaliya districts. The district covers an area of about 311.7 km$^2$, with land use distribution consisting of approximately 57% cultivation, 17% grazing, 26% forests, and 1% allocated for other purposes. The district experiences a mean annual temperature ranging from 10˚C to 25˚C, and annual rainfall ranges from 900 to 1800 mm. The altitude in the area varies between 1100 and 2900 meters above sea level (m.a.s.l). The district was categorized into three agroecological zones based on altitude and temperature: highland (44%), midland (31%), and lowland (25%), consisting of eight, five, and three kebeles, respectively. Roge Danisa and Roge Ajo from the highland, Liban Gamo from the midland, and Haro Marami from the lowland areas, in four kebeles were purposively selected based on their potential for indigenous chicken production [25].

### Data collection and sample size

Data collection involved field visits to the district and target sampling kebeles, where indigenous chickens were observed and measured. The sample population was selected using purposive sampling. The sample size of the household for the study was determined according to the [26] formula, which is given by N $= \frac{0.25}{SE^2}$, where N represents the sample size and SE denotes the standard error of the sample size.

To determine the sample size, households with at least two matured indigenous chickens (hens and/or cocks) was listed. Thus, a total of 2166 households and 2452 chickens were surveyed from which, a total of 192 households and 224 chickens were selected, and parametric data were collected. In this study, a standard error of 0.05 was used with a 95% confidence level. Following the phenotypic characterization guidelines provided by [18], it was recommended to include 100–300 matured females and 10–30 matured males to maintain validity. Therefore, a total of 224 (56 from each kebele) matured indigenous chickens were sampled, consisting of 138 females and 86 males from which both qualitative and quantitative measurements were taken. Qualitative traits, which are observable characteristics that do not involve numerical measurement, were collected through visual observation of traits such as head shape, comb type, feather plumage, shank color, earlobe color, and skin color. Whereas, quantitative measurements for the parameters like body weight and various linear body dimensions that included body length, body width, shank length, shank circumference, and wingspan were taken using a flexible tape meter.

The socio-economic characteristics of the respondents, including their age, gender, household size, education level, and primary occupation, were assessed across the three agroecological zones of the study area. A structured questionnaire was used to collect data on these variables, as well as the respondents' roles and responsibilities in chicken management. A multi-stage sampling approach was employed, first stratifying the study area into the three agroecological zones, then randomly selecting representative villages within each zone, and finally randomly choosing respondents from the selected villages.

The Food and Agriculture Organization [27] was used as comprehensive guidelines for appropriate sampling strategies, standardized data collection protocols, and recommended statistical analysis methods, phenotypic characterization, and assessing a combination of qualitative traits (plumage color, comb type, shank color) and quantitative traits (body weight, body length, egg production).

## Data analysis

The collected data was organized, cleaned, and processed to ensure accuracy and usability and analyzed using the Statistical Package for Social Sciences (SPSS Ver.20) considering variables such as sex and agroecology as independent variables. These software packages offer a range of statistical techniques and tools to explore relationships, test hypotheses, and generate results.

Yij = μ + Ai +Bj +ABij +eij Where

Yijk: the corresponding quantitative trait of local chicken in ith agro-ecology (i = 3, highland, midland & lowland) of jth sex (j = 2, male and female)

μ: overall population mean for the corresponding quantitative trait

Ai: effect of i[th] agro-ecology

Bj: effect of j[th] sex (j = 2, male & female)

ABij: agro-ecology & sex interaction effect

eij: residual error [28]

Descriptive statistics tests, such as chi-square and ANOVA, were used to analyze the data and identify any significant differences in socio-economic characteristics and chicken management roles between the agroecological zones and the degree of sexual dimorphism in the traits across the different agroecological contexts. As well, descriptive statistics also used to generate insight into the phenotypic characteristics of the indigenous chicken population in the study area. Pearson correlation provides insights into the relationships between body weight (BW) and various linear body measurements of the chicken study area. The Chi-Square Tests was employed to assess the associations between categorical variables, such as qualitative traits and agroecological zones. The ANOVA (Analysis of Variance) was used to determine if there are statistically significant differences between the means of different groups, particularly when examining quantitative traits. And, the Correlation Analyses were conducted to evaluate the relationships between various quantitative traits, such as shank length and body weight.

In addition, PCA (Principal Component Analysis) was run to reduce dimensionality and identify the main factors influencing phenotypic variation within the indigenous chicken population, visualize the relationships among traits and reveal the underlying structure of the data, supporting further insights into the phenotypic diversity present.

## Results and discussion

### General socio-economic characteristics

The socio-economic characteristics provide insights into the demographic composition of the respondents and their roles in chicken management within the different agroecologies of the study area. The socio-economic characteristics across the three agroecologies of this study are presented in Table 1. Generally, the socioeconomic characteristics highlighted the limited education among respondents, the lack of awareness of proper chicken management, and the prevalence of Protestantism in the study area. The majority of the respondents in all three agroecologies were females (68.8%) and this finding is consistent with [29] that reported 70% are female poultry keepers and this suggests that females are the primary role players compared to males. The predominance of female respondents in smallholder chicken production is attributable to cultural norms positioning it as a "woman's domain," women's availability for daily management tasks, and their access to and control over chickens as "women's assets" that can generate household income [30,31].

Of all the respondents, around 26.6% and about 11% fall within the age groups of 31–40 and 51–65 years, respectively, highlighting the presence of a productive age group for effective

**Table 1. Respondents and owners' profiles in the study area.**

| Agroecological zone | | | | |
|---|---|---|---|---|
| Respondent profile | High altitude n = 96 | Mid altitude n = 48 | Low altitude n = 48 | Total n = 192 |
| **Sex** | | | | |
| Male | 24 (25) | 20 (41.7) | 16 (33.3) | 60 (31.2) |
| Female | 72 (75) | 28 (39.3) | 32 (56.7) | 132 (68.8) |
| **Age of respondent** | | | | |
| <20 | 18 (18.8) | 11 (22.9) | 11 (22.9) | 40 (20.8) |
| 20–30 | 18 (18.8) | 12 (25) | 12 (25) | 42 (21.9) |
| 31–40 | 30 (31.2) | 10 (20.8) | 11 (22.9) | 51 (26.6) |
| 41–50 | 19 (19.7) | 13 (27.1) | 6 (12.5) | 38 (19.8) |
| 51–65 | 11(11.5) | 2 (4.2) | 8 (16.7) | 21 (10.9) |
| Educational level of respondents | | | | |
| Unschooled | 12 (11.5) | 5 (10.4) | 6 (12.5) | 23 (12) |
| Religious school | 14 (14.6) | 11 (22.9) | 3 (6.2) | 28 (14.6) |
| Writing and reading | 34 (35.5) | 16 (33.3) | 20 (41.7) | 70 (35.4) |
| Primary(1–8) | 24 (25) | 10 (20.8) | 11 (22.9) | 45 (24) |
| High school | 12 (12.5) | 6 (12.6) | 8 (16.7) | 26 (14) |
| **Farming systems** | | | | |
| Crop production | 3(3.2) | 5(10.4) | 3(6.3) | 11(5.7) |
| Livestock production | 7(7.3) | 2(4.2) | 2(4.2) | 11(5.7) |
| Mixed farming systems **Religious of respondents** | 86(89.5) | 41(85.4) | 43(89.5) | 170(88.6) |
| Protestant | 55(57.3) | 31(64.6) | 24(60.4) | 110(57.4) |
| Orthodox | 31(32.3) | 13(27.1) | 17(35.4 | 61(31.7) |
| Waqeffataa | 10(10.4) | 4(8.3) | 2(4.4) | 21(10.9) |
| Family size/HH (Mean ± S E) | 4.52±0.16 | 4.81 ±0.17 | 5.25±0.21 | 4.65±0.18 ns |
| Landsize/HH (ha.) (Mean ± SE) | 1.54±0.12 | 1.73±0.19 | 2.34±0.22 | 1.87±0.17 |

n = Number of respondents = significance, number in the parentheses show % of respondent.

chicken management. Age affects the level of experience, knowledge, and ability to adapt to new farming practices [32].

Respondents' educational levels varied across the agroecological zones, with the highest proportion having completed primary education in the high-altitude (47%) and mid-altitude (52%) zones, while the majority (48%) in the low-altitude zone had no formal education. This variation in educational attainment aligns with findings that smallholder farmers' access to education can differ based on their geographical location, which may influence their adoption of improved chicken management practices, as more educated farmers tend to be more receptive to new technologies and information [33].

The educational level plays an important role in accessing information, adopting modern techniques, and making informed decisions related to poultry management, disease control, and market trends. Higher levels of education are often associated with better farm management practices and increased productivity [32].

The majority of respondents (88.6%) were engaged in mixed crop-livestock production systems and kept chickens for immediate income generation to cover various expenses, such as purchasing salt, coffee, clothes, and medication. Similar findings have been reported in other parts of Ethiopia [19,34,35].

In terms of religion, 57.3% of the surveyed population identified as Protestant, while 31.7% were Orthodox and 10.9% were Waqefataa, indicating more Protestant habituating in the study area and religion could influence poultry meat consumption [36].

The family size per household did not show any significant differences (p < 0.05) between the highland and midland agroecologies with an overall mean of 4.65±0.18, it is established that household size impacts the available labor force and resources for investment in poultry production and influences the income earned from poultry farming [32].

The study found that the mean family size in the study area was 4.65±0.18, which is similar to the average of 5.5±0.22 reported for the Oromia region [37]. The average land size per household was 1.87±0.17 hectares, indicating sufficient land for agricultural activities. Notably, lowland areas had the largest average land size per household (2.34±0.22ha), while highland areas had the smallest (1.54±0.12ha), with a significant difference (p<0.05). However, there was no significant difference (p > 0.05) between midland and the other two agroecologies. The land holding size in the area was considerably larger than the 0.86 ha reported in the Dale, Wonsho, and Loka Abaya districts of South Ethiopia [38], that reported land holding size of 0.86 ha in the South Ethiopian districts, but consistent with the national average of 1.18 ha [37], and correlated closely with population density [39]. Highland and midland agroecologies were found to be more suitable for both crop and animal production, while lowland areas faced challenges due to limited animal feed caused by low rainfall [39].

The study's findings reveal that the socio-economic characteristics of the farming households and communities in the Liben Jawi district, beyond just landholdings and market orientation, are closely associated with the observed phenotypic diversity of the indigenous chicken populations. The chickens belonging to households headed by older farmers tended to exhibit more traditional plumage patterns and comb types, suggesting that age and traditional knowledge influence the conservation of certain cultural traits within the local chicken ecotypes and this is in line with [40], Similarly, the educational level of the household heads was positively correlated with the prevalence of improved production traits, indicating that more educated farmers are better able to implement management practices that support the expression of these economically valuable characteristics [41].

## Phenotypic characterization

**The qualitative trait of indigenous chickens.** The current study found that the indigenous chickens in the study area were all normally feathered, with 100% of females and 100% of males exhibiting this trait (Table 2). This finding is consistent with the previous research conducted by [21], who also observed that most indigenous chicken ecotypes were normally feathered. Additionally, a majority of the indigenous chickens in the study area had no feathered shanks, with only 33.7% of hens and 49.3% of cocks displaying feathered shanks. This observation aligns with the findings of [8,42,43], who reported that most indigenous chicken ecotypes lack shank feathers and have yellowish-colored shanks. Similar results have been reported in more recent studies on the phenotypic characterization of indigenous chickens in different parts of Ethiopia. Similarly [44] found that most indigenous chickens in the Metekel zone were normally feathered, with only a small proportion having feathered shanks. Likewise, [45] observed that indigenous chickens in the Bale highlands were predominantly normally feathered, with a low prevalence of feathered shanks.

Table 2 and Fig 3 illustrate the diverse plumage coloration observed in the chicken population. The results indicate that the dominant colors for hens/females in the study area were brown mottled (23.9%), red (23.1%), black (9.4%), dark brown (7.9%), black laced white (6.5%), reddish brown (5.2%), and greyish mixture (5.1%). These findings differ from the

**Table 2. Summary of PCA, cluster analysis, and body index values for indigenous chickens.**

| Analysis Type | Metric/Characteristic | Males (N = 112) | Females (N = 112) | Overall (N = 224) |
|---|---|---|---|---|
| PCA Results | PC1 Variance Explained (%) | 35.4 | | |
| | PC2 Variance Explained (%) | 25.1 | | |
| | PC3 Variance Explained (%) | 15.2 | | |
| | PC4 Variance Explained (%) | 10.5 | | |
| | PC5 Variance Explained (%) | 8.1 | | |
| | PC6 Variance Explained (%) | 5.7 | | |
| Cluster Analysis | Cluster 1 Traits | Plain head shape, single comb, red plumage | | 80 |
| | Cluster 2 Traits | Domed head shape, rose comb, brown mottled | | 50 |
| | Cluster 3 Traits | Angular head shape, pea comb, black feathers | | 40 |
| | Cluster 4 Traits | Mixed traits | | 54 |
| Body Index Values | Average Body Weight (kg) | 1.46 ± 0.02 | 1.21 ± 0.01 | 1.34 ± 0.015 |
| | Average Body Length (cm) | 38.05 ± 0.26 | 31.55 ± 0.33 | 34.80 ± 0.30 |
| | Average Body Width (cm) | 22.20 ± 0.20 | 19.04 ± 0.21 | 20.62 ± 0.21 |
| | Average Shank Length (cm) | 8.76 ± 0.11 | 6.66 ± 0.05 | 7.71 ± 0.08 |
| | Average Wingspan (cm) | 38.56 ± 0.13 | 32.98 ± 0.13 | 35.77 ± 0.14 |

Note: *PC Variance Explained (%)" refers to the percentage of the total variance in the dataset that is captured by each principal component (PC) in Principal Component Analysis (PCA).*

reports of [46], who documented white, black, red, and black with white stripes as the prevalent plumage colors in North West Ethiopia. The wide variation in plumage coloration could be attributed to a lack of selection for specific traits in the local chicken population, a phenomenon also observed in more recent studies. For instance, [47,48] reported a diverse range of plumage colors in indigenous chickens in the Amhara region of Ethiopia, including brown, white, black, and multicolor patterns. Similarly, [10,45] found a high diversity of plumage colors in indigenous chickens in the Bale highlands, with the most common colors being brown, black, and white. This wide variation in plumage coloration is a characteristic of indigenous chicken populations that have not been subjected to targeted breeding programs, as observed in recent studies from Nigeria [49]. For cocks/males, the dominant plumage colors were red (26.7%), brown mottled (19.8%), dark brown (9.3%), black laced white (8.1%), brown mottled (5.8%), reddish brown (5.6%), and greyish mixture (4.7%). Also [50] reported a wide range of plumage coloration in indigenous chicken populations in the Bale highlands of Ethiopia, highlighting the adaptability and survival benefits of diverse plumage colors, which can be attributed to the absence of targeted selection breeding programs for specific colors, similar to the observations in Nigerian native chickens by [49].

Red was the predominant plumage color in male chickens, accounting for approximately 25.1% in highland, 33.3% in midland, and 23.8% in lowland agroecologies. This finding is consistent with recent studies highlighting the prevalence of red plumage in indigenous chickens in Ethiopia. According to [45] red was the most common plumage color in male indigenous chickens in the Amhara region, accounting for 27.3% of the population. Similarly, [51] observed a high proportion of red plumage in male indigenous chickens in the Bale highlands of Ethiopia, with red accounting for 32.5% of the plumage colors. These consistent findings across multiple recent studies suggest that red plumage is a defining feature of male indigenous chickens in the country.

Black, white, and yellowish-brown plumage colors were also present in the studied populations, but in lower proportions compared to the predominant red plumage. This is in line with

observations from other recent studies on indigenous chickens in Ethiopia [7,52,53] and across Africa [54,55], which have reported the co-existence of various plumage colors, with red being the most common. The presence of a black breast, which was almost absent in females, is a notable sexual dimorphism that has been documented in various recent studies on indigenous chicken populations [7,51].

Univariate analysis revealed distinct qualitative trait distributions between male and female indigenous chickens. In terms of comb types, single comb was the predominant type in males, accounting for 39.5%, followed by pea comb at 37.2%. Among females, 75.4% had single combs, with pea combs being the second most common at 22.5%. These results align with the findings of [56] in the Gurage zone of southwestern Ethiopia, where single comb was the predominant type in both males (58.2%) and females (70.1%). Similar patterns of comb-type distribution have been observed in other recent studies on indigenous chickens in Ethiopia [53,56] and across Africa [54,55], Quantitative traits also exhibited significant sex-based variations. Cocks were heavier (1.46 ± 0.02 kg) and had larger body lengths (38.05 ± 0.26 cm) and wingspans (38.56 ± 0.13 cm) compared to hens (1.21 ± 0.01 kg, 31.55 ± 0.33 cm, 32.98 ± 0.13 cm), suggesting that single and pea comb types are characteristic of indigenous chicken populations in the region [1]. The multivariate analysis, including principal component analysis (PCA) and cluster analysis, further explored the underlying patterns and relationships within the phenotypic data. The PCA likely identified the main sources of variation and the correlations between the different traits, providing insights into the key factors contributing to the observed phenotypic diversity [1]. The cluster analysis grouped the chickens based on their overall phenotypic characteristics, potentially revealing distinct ecotypes or subpopulations within the indigenous chicken population (Table 2). The correlation and regression analyses explored the relationship between shank length and body weight, indicating a strong positive correlation (r = 0.45 for hens and r = 0.44 for cocks, p < 0.001), suggesting that shank length could serve as a reliable predictor of body weight, which has practical implications for on-farm phenotypic assessments and breeding programs [1].

The calculated Body Weight Index (BWI) revealed that hens had a higher BWI of 0.0741 compared to cocks at 0.0576, suggesting better body condition or muscularity in hens (Melesse, 2014). The Comb Index (CI) showed that cocks had a lower CI of 2.02 compared to hens at 2.14, indicating relatively larger and more pronounced combs in cocks, a secondary sexual characteristic (Melesse, 2014). Similarly, the Wattle Index (WI) was higher in cocks at 1.77 compared to hens at 1.64, suggesting cocks had relatively longer and narrower wattles, another sexually dimorphic trait [1] (Figs 1 and 2).

The PCA scatter plot of the indigenous chickens' morphological indices (Fig 3) reveals significant phenotypic diversity within the studied population [57]. The identification of three distinct and one mixed clusters (Fig 3) suggests the presence of adaptive strategies and variations in physical characteristics among the chickens [58].

Cluster 1, characterized by larger body sizes and more robust physical features, may confer advantages in foraging efficiency and overall fitness [59] These chickens could be more desirable for farmers due to their increased productivity and resilience [60]. In contrast, Clusters 2 and 3, with smaller body sizes, may exhibit different adaptive strategies influenced by factors such as feed availability, disease resistance, or thermal regulation requirements [61]. The observed correlations between morphological traits within each cluster provide valuable insights into the underlying relationships and the potential for targeted selection [62]. Selecting for increased body weight may lead to correlated improvements in other economically important traits, such as growth rate and egg production [58].

The study found significant differences in head shape between male and female chickens (p<0.01) as well as among the studied agroecologies (p<0.05). Approximately 54.7% of males

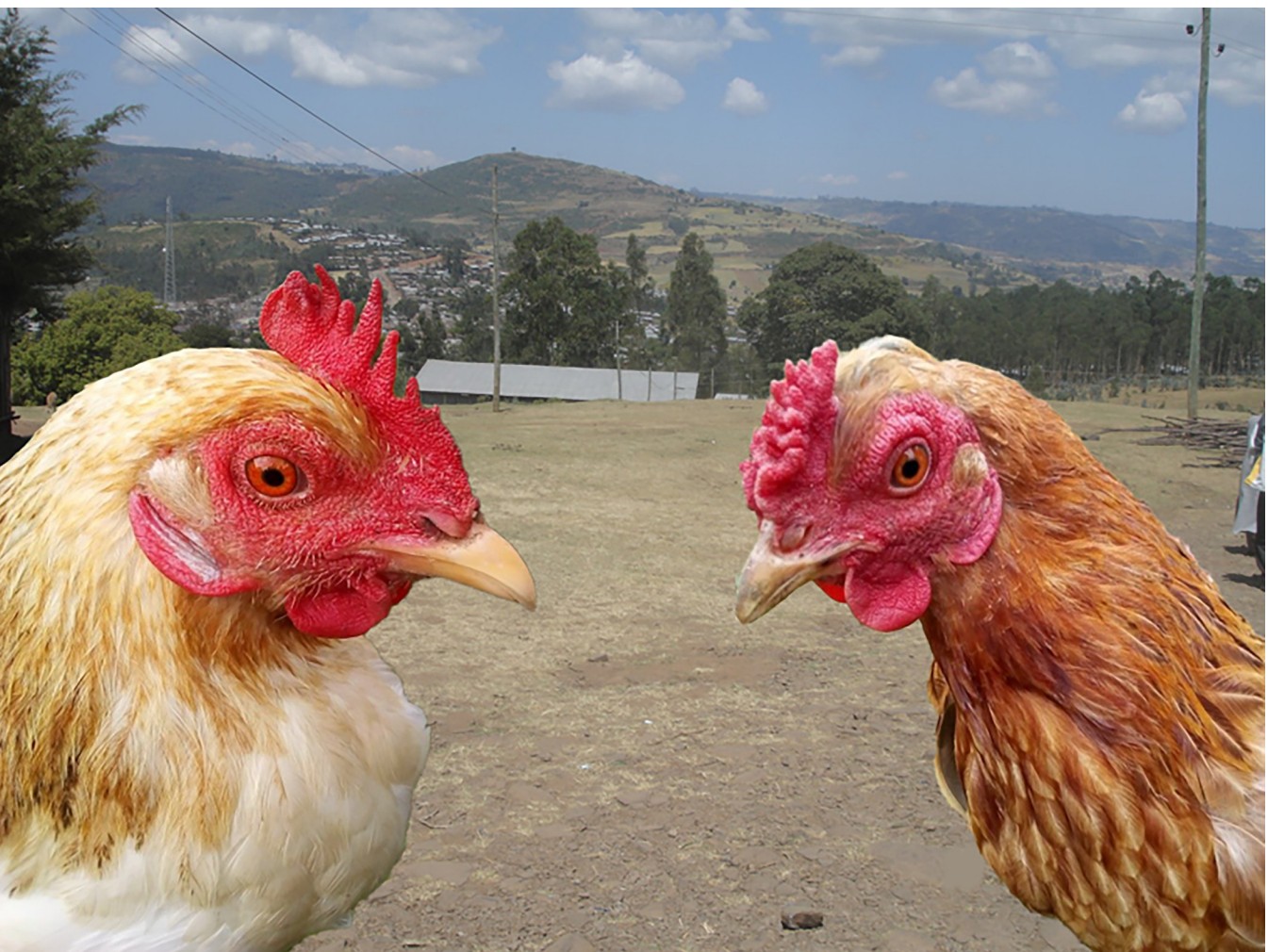

**Fig 1. Variations of comb type among Indigenous chickens.**

and 59.4% of females had a plain head shape, with no significant difference in plain head shape among the agroecologies. This contrasts with the earlier findings of [21], who observed crest heads were predominant in the northern population and plain heads were characteristic of the southern population. This discrepancy may be attributed to breed differences; as more recent studies have highlighted the diversity of head shapes within indigenous chicken populations in Ethiopia.

A study by [51] on indigenous chickens in the Bale highlands found that plain head was the most common head shape, accounting for 58.4% of the population. Similarly, [16] reported that plain head was the predominant head shape in indigenous chickens in the Amhara region, observed in 58.2% of males and 70.1% of females. These findings align with the current study, suggesting that plain head shape is a common characteristic of indigenous chickens across different regions of Ethiopia.

Regarding shank color, the study found that yellow was the predominant shank color in both males (54.5% in highland, 47.6% in midland, and 61.9% in lowland) and females (50.7% in highland, 54.3% in midland, and 44.4% in lowland) chickens, with significant differences ($p < 0.05$) among the agroecologies. These results are consistent with more recent studies on

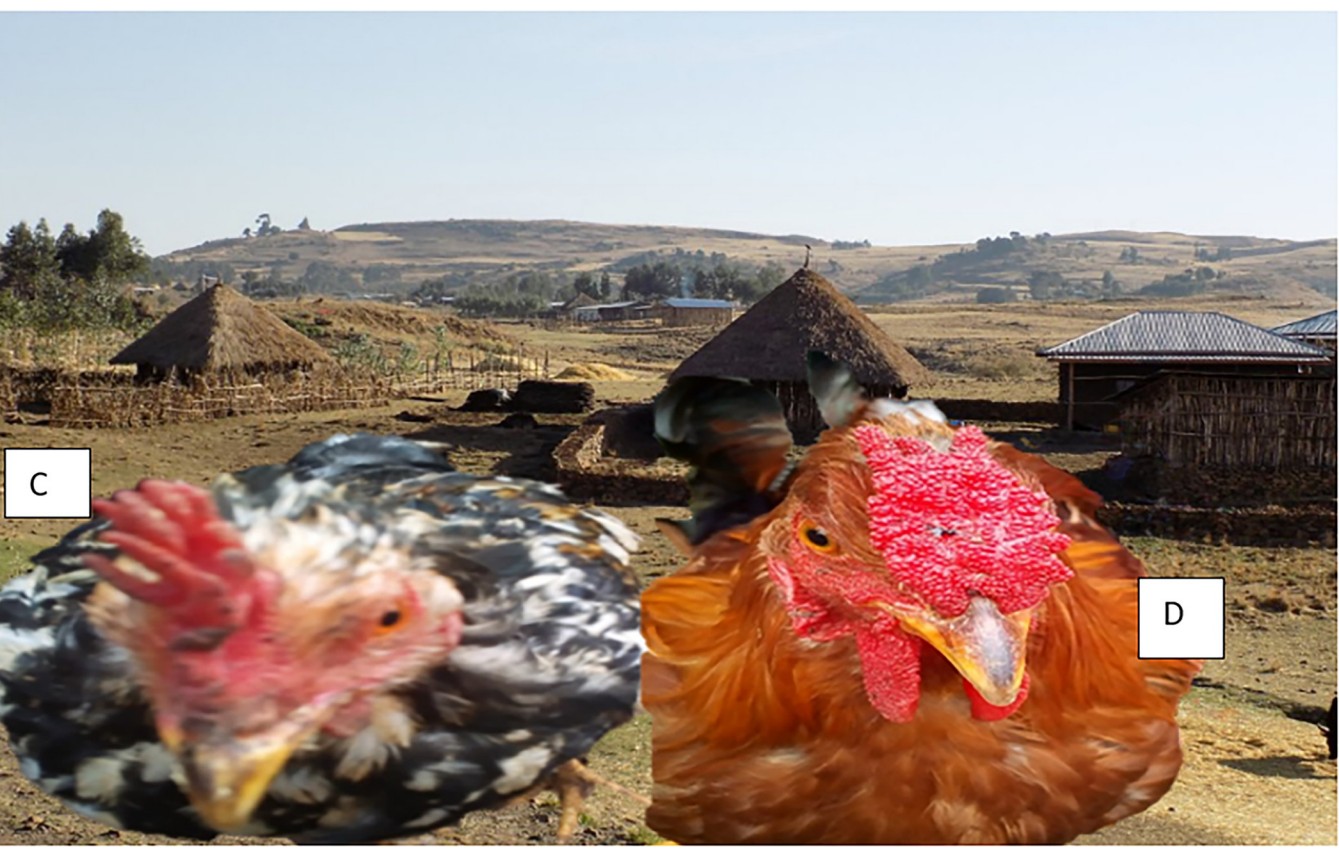

**Fig 2. Variations of comb type among Indigenous chickens.**

indigenous chickens in Ethiopia. [7] reported that yellow was the dominant shank color in indigenous chickens in the Bench Sheko Zone, accounting for 56.3% of the population. Similarly, [53] found that yellow shank color was the most common in indigenous chickens in the central highlands of Ethiopia, observed in 55.6% of the birds.

The prevalence of yellow shank color in indigenous chickens in Ethiopia has been well-documented in the literature. The current findings align with the earlier reports by [10,61], further confirming the dominance of yellow shank color as a characteristic of indigenous chicken populations in the country.

For earlobe color, the study found that red earlobes were dominant in both males (51.6%) and females (51.4%) (Fig 4)., with significant differences (p < 0.03) among the agroecologies (Table 2). Recent studies on indigenous chickens in Ethiopia have reported the prevalence of red earlobe color as the most common earlobe color in indigenous chickens in the Bale highlands, accounting for 57.3% of the population [51], and red earlobe was the dominant type in indigenous chickens in the Amhara region, observed in 52.7% of the birds [16].

These consistent findings from multiple studies suggest that red earlobe color is a characteristic feature of indigenous chickens in Ethiopia, although the proportions may vary across different regions and populations. The discrepancy between the current study and the earlier reports may be attributed to regional or breed-specific differences in the expression of this trait [16,63].

The study found significant variation in plumage color and earlobe color among the indigenous chickens, with diverse phenotypes including brown earlobe with black plumage and pea

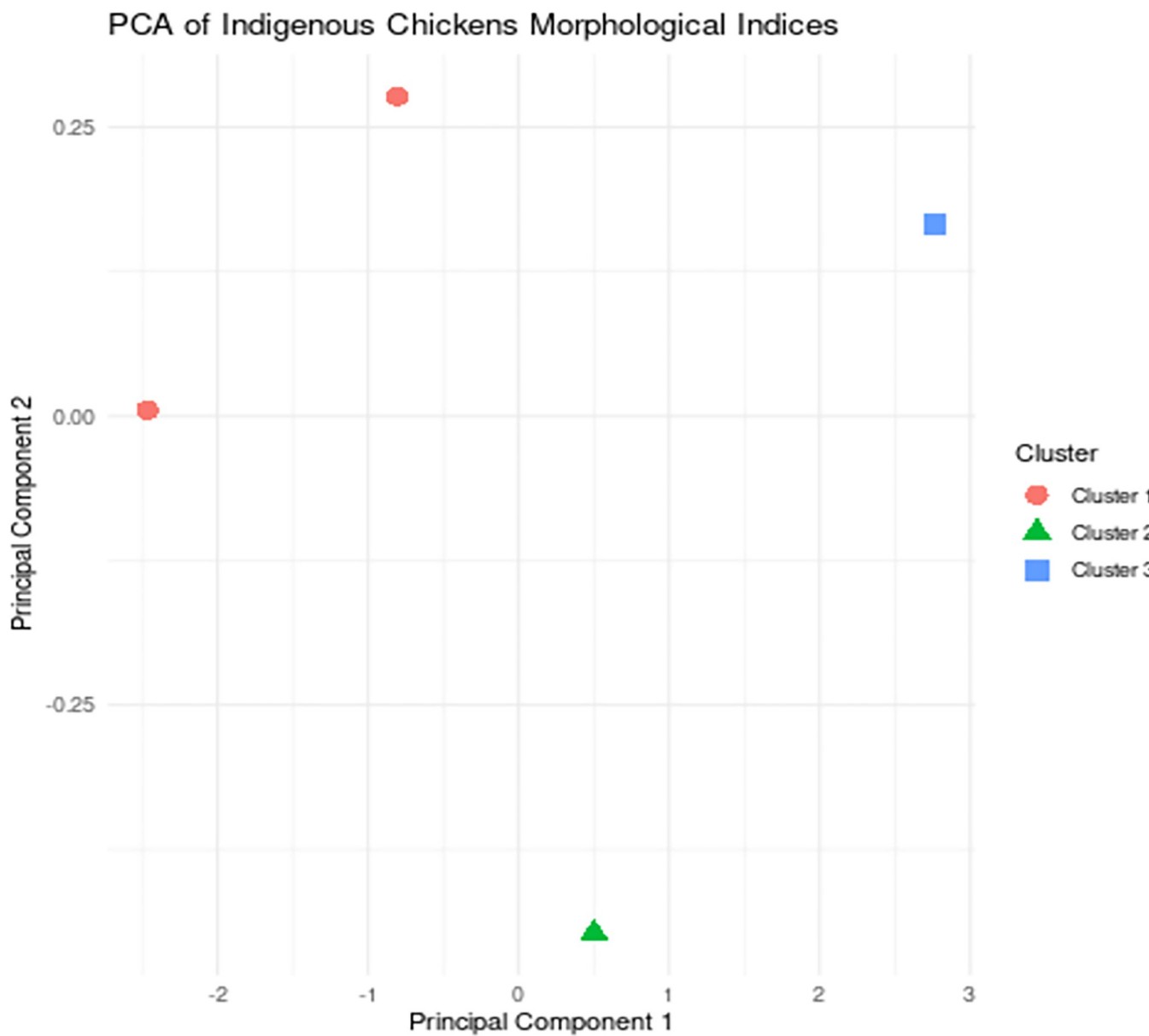

**Fig 3. Indigenous chickens' morphological indices based on Principal Component Analysis (PCA).**

comb, as well as white plumage with red earlobe and single comb (Fig 5). This diversity aligns with previous research [21] and has been further corroborated by more recent studies documenting significant differences in trait frequencies among indigenous chicken ecotypes in Ethiopia [45,58]. The rich phenotypic diversity is attributed to these indigenous chickens' genetic heterogeneity and adaptive capacity, which have evolved under diverse agroecological conditions and traditional farming practices [10,64,65].

The study results on the qualitative traits of indigenous chickens across three different agroecological zones: highland, midland, and lowland showed notable differences in the phenotypic characteristics of the chickens between the zones (Table 2).

There are significant differences in plumage color distributions across the agroecological zones (p = 0.009). The highland zone had the highest percentage of black-mottled (6.8%) and

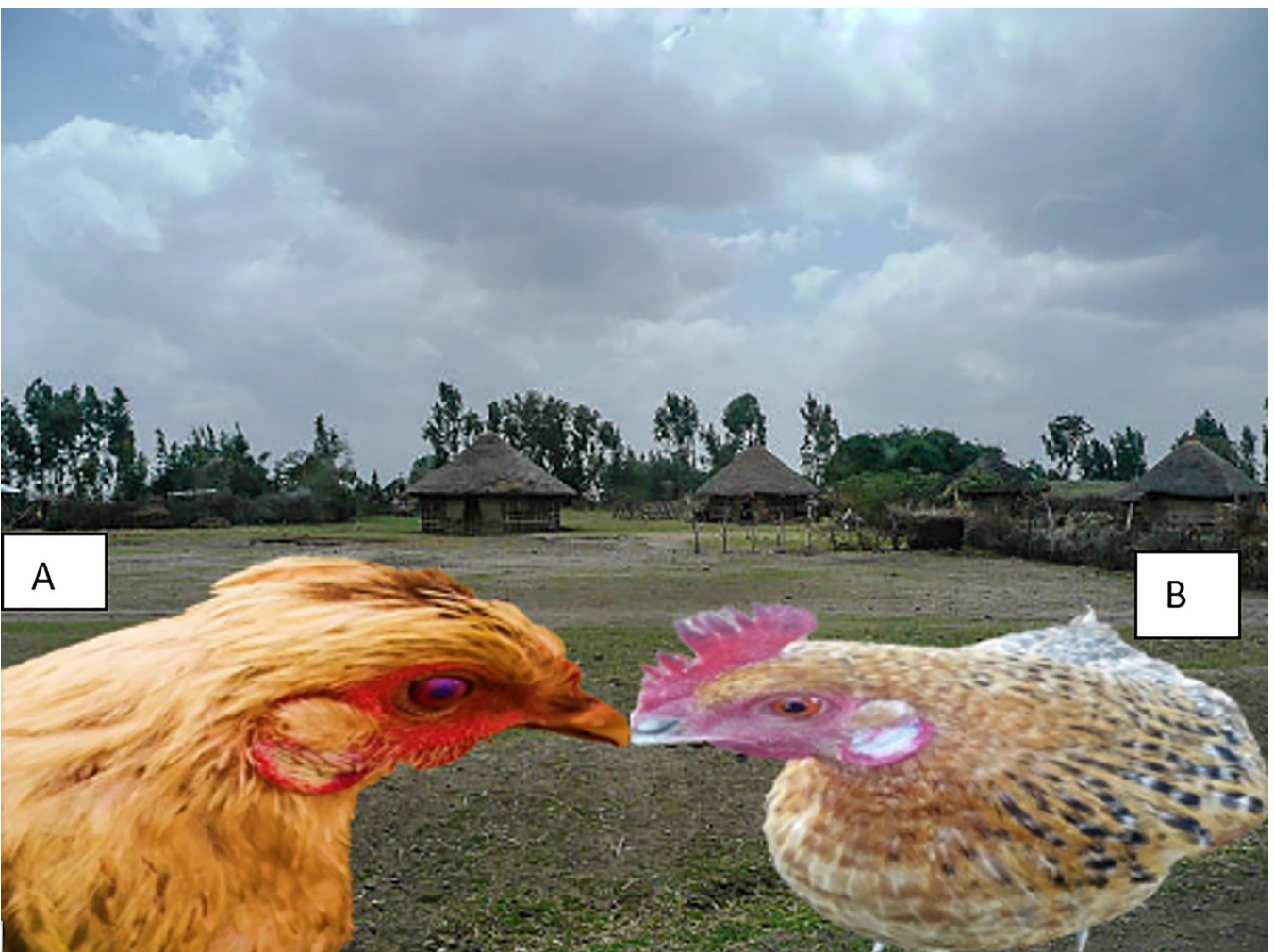

**Fig 4.** White ear lobe and brown plumage color hen (A), and white ear lobe and black-laced white plumage color hen/female/ (B).

black-laced white (6.6%) plumage colors among cocks, while the lowland zone had the highest percentage of black-laced white (19.1%) plumage in cocks. The midland and lowland zones also had higher proportions of brown mottled plumage in both cocks and hens compared to the highland zone. These findings are consistent with recent studies suggesting that plumage color in chickens can be influenced by environmental factors such as temperature and solar radiation [66,67], the higher prevalence of darker plumage colors in the highland zone may be an adaptation to the cooler temperatures and higher UV exposure in that environment, as melanin-based pigmentation can provide better protection against solar radiation [68].

The distribution of comb types did not differ significantly between the zones (p = 0.608). The single comb was the predominant type, observed in 34.1–82.4% of the chickens, followed by the pea comb in 17.6–38.6% of the birds. This finding suggests that the comb type distribution was relatively consistent across the different zones, despite potential variations in other factors. Comb type is a heritable trait in chickens, and its distribution can be influenced by factors such as breed, genetics, and environmental conditions [69].

There were significant differences in head type distribution across the zones (p = 0.017). The highland zone had the highest percentage of chickens with crested heads (47.7% of cocks,

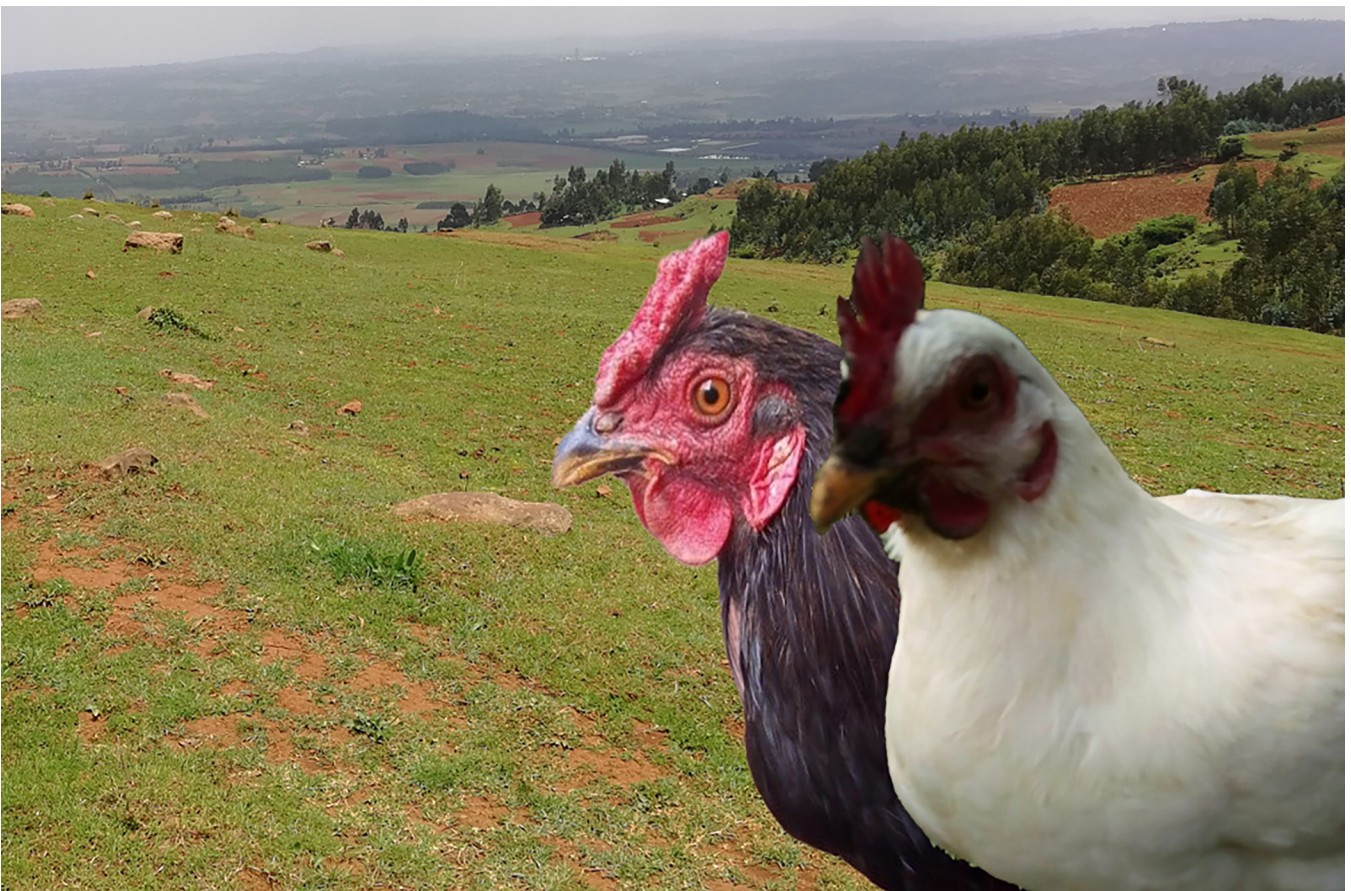

**Fig 5. Shows inconsistency in plumage color and ear lobe.** (A), Brown ear lobe, black plumage color, pea combed type, and red eye color, (B), white plumage color, red ear lobe, and eye with single comp type).

44.9% of hens), while the lowland zone had the lowest (38.1% of cocks, 32.4% of hens). The presence of crested heads may provide better protection against the elements in the highland environment, as suggested by previous studies on the adaptive significance of head morphology in chickens [70].

Significant differences were also observed for ear lobe color (p = 0.037), eye color (p = 0.007), and shank color (p = 0.028) between the agroecological zones. These variations could be related to factors such as thermoregulation, disease resistance, or other fitness-related characteristics [71,72].

The observed differences in qualitative traits of indigenous chickens across the agroecological zones suggest that the local environment plays an important role in shaping the phenotypic diversity of these populations [73–75].

These findings highlight the importance of considering the local environmental context when characterizing and conserving indigenous chicken genetic resources.

The performed chi-square tests statistical analysis indicated the degree of sexual dimorphism in the traits across the different agroecological contexts in which the proportions between cocks and hens were compared. The earlobe color, showed no significant differences in the proportion of cocks and hens with red earlobes in the highland ($\chi^2(1) = 0.01$, p = 0.92), lowland ($\chi^2(1) = 0.43$, p = 0.51), and midland ($\chi^2(1) = 2.48$, p = 0.12), indicating a relatively consistent degree of sexual dimorphism in this trait across the different environments.

Similarly, for head type, there were no significant differences in the proportion of cocks and hens with crest head type in the highland ($\chi^2(1) = 0.35$, p = 0.55), midland ($\chi^2(1) = 1.02$, p = 0.31), and lowland ($\chi^2(1) = 0.61$, p = 0.43) contexts, suggesting that the degree of sexual dimorphism in head type was also relatively consistent across the agroecological contexts. According to these findings, the degree of sexual dimorphism in earlobe color and head type may be largely influenced by other factors like genetics and less influenced by the environmental factors associated with the different agroecological contexts [76,77]. In contrast, the degree of sexual dimorphism in comb type and shank feather presence may be more influenced by environmental factors, as these traits can be more responsive to differences in resource availability, climate, and other ecological conditions [78].

The interaction between AE and sex also plays a crucial role in influencing these traits and further ascertains, that for traits where AE is significant, the interaction with sex is also noted as significant, suggesting that the impact of AE may vary depending on the Sex of the chickens [79–81].

The chi-square tests of independence revealed several statistically significant associations between the examined plumage and qualitative traits in the study population. Plumage color, head type, ear lobe color, eye color, shank color, and skin color were all found to be significantly related to other variables, suggesting these traits may be influenced by or correlated with various genetic, environmental, or behavioral factors [82]. In contrast, comb type and shank feather were not significantly associated with the other traits, potentially indicating their independence or unique determinants [83]. These findings align with previous research on the complex relationships between phenotypic characteristics in avian species, where multiple interacting factors can contribute to the expression of morphological diversity [72].

## Quantitative traits

The mean body weight values were found to be 1.46±0.027 kg for males and 1.12±0.012 kg for females, with significant differences (p<0.05) observed across all agroecologies for both sexes (Table 3). These results align with earlier reports by [45,84,85], who also documented similar mean body weights for indigenous chickens in Ethiopia.

However, the current findings differ from the observations of [86] in the Guji Zone, who reported higher body weights of 2.1±0.05 kg for males and 1.5±0.02 kg for females. These variations in body weight may be attributed to differences in the genetic background of local chicken populations, management practices, and the availability of feed resources in different regions [58].

Sexual dimorphism, with males (cocks) weighing higher than hens, was observed in the current study, which is consistent with the findings reported by [21,61]. This difference in body weight between sexes is a common characteristic of indigenous chicken populations [51,85].

The average body width and body length of indigenous chickens were measured to be 22.20 ±0.20 cm and 19.04±0.21 cm, respectively, in males, and 38.05±0.269 cm and 31.55±0.33 cm, respectively, in females. These measurements showed significant differences (p<0.05) across all agroecologies for both sexes, reflecting the phenotypic diversity within the local chicken populations.

The average shank length in males was 8.76±0.110 cm, which was slightly lower than the values reported by Halima et al. (2007) in Northwestern Ethiopia (8.91 cm) and by [10] in Southern Ethiopia (9.1 cm). The shank circumference values were measured to be 3.58±0.05 cm in males and 3.36±0.03 cm in females, consistent with the findings for Wareng-Tangerang Indonesian local chickens [87].

The findings of the current study on the body weight and linear body measurements of indigenous chickens in the study area are consistent with and expand upon previous research

**Table 3. Qualitative traits of indigenous chicken in the study areas.**

| Agroecology | | | | | | | | | | |
|---|---|---|---|---|---|---|---|---|---|---|
| **Qualitative trait** | **Highland** | | **Midland** | | **Lowland** | | **Overall** | | $\chi^2$ | **p-value** |
| | Cock% | Hen = % | Cock = % | Hen = % | Cock = % | Hen = % | Cock | Hen % | | |
| Plumage Color | | | | | | | | | 42.78 | 0.009 |
| Black | 6.8 | 10.1 | 4.8 | 5.7 | 4.8 | 4.2 | 5.8 | 9.4 | | |
| Black mottled | 6.8 | 4.3 | No | No | No | 5.7 | 3.5 | 3.6 | | |
| Black-laced white | 6.6 | 4.3 | No | 8.6 | 19.1 | 8.6 | 8.1 | 6.5 | | |
| Brown | 9.1 | 1.4 | No | 5.7 | 3.5 | 5.7 | 5.8 | 3.6 | | |
| Brown mottled | 18.2 | 24.6 | 14.3 | 28.1 | 28.6 | 14.3 | 19.8 | 23.9 | | |
| Dark brown | 6.7 | 8.7 | 19 | 2.9 | 2.3 | 11.4 | 9.3 | 7.9 | | |
| Dark Brown mottled | 7.1 | 4.3 | 9.5 | 2.1 | No | 2.9 | 5.8 | 3.6 | | |
| Greyish mixture | 0 | 5.8 | 17 | 2.7 | No | 5.7 | 4.7 | 5.1 | | |
| Red | 25.1 | 15.9 | 31.1 | 31.4 | 23.8 | 28.6 | 26.7 | 23.1 | | |
| Reddish brown | 4.5 | 10.1 | No | 1.4 | 14.3 | No | 5.6 | 5.2 | | |
| Wheaten | 0 | 1.4 | No | No | 1.3 | No | No | 3.6 | | |
| Wheaten mottled | 2.3 | No | No | 4.2 | No | 1.2 | 1.4 | 1 | | |
| White | 2.3 | 5.8 | 4.3 | 4.1 | 2.3 | 6 | 1.2 | 1.2 | | |
| White mottled | 4.5 | 3.3 | No | 3.1 | No | 5.7 | 2.3 | 2.3 | | |
| Comb Type | | | | | | | | | 4.44 | 0.608 |
| Single | 34.1 | 72.5 | 38.1 | 74.3 | 52.4 | 82.4 | 39.5 | 75.4 | | |
| Pea | 38.6 | 24.6 | 38.1 | 22.9 | 33.3 | 17.6 | 37.2 | 22.5 | | |
| Rose | 13.6 | 2.9 | 14.3 | 2.8 | No | No | 10.5 | 2.1 | | |
| Double | 13.7 | No | 9.5 | No | 14.3 | No | 12.8 | No | | |
| Head Type | | | | | | | | | 12.11 | 0.017 |
| Crest | 47.7 | 44.9 | 47.6 | 40.0 | 38.1 | 32.4 | 45.3 | 40.6 | | |
| Plains | 52.3 | 55.1 | 52.4 | 60.0 | 61.9 | 67.6 | 54.7 | 59.4 | | |
| Ear lobe Color | | | | | | | | | 14.45 | 0.037 |
| Red | 54.5 | 53.6 | 38.1 | 48.6 | 57.1 | 50 | 51.6 | 51.4 | | |
| White | 43.2 | 31.9 | 33.3 | 28.6 | 42.9 | 32.4 | 40.7 | 31.2 | | |
| Yellow | 2.3 | 2.9 | 14.3 | No | No | 8.8 | 4.7 | 3.6 | | |
| Brown | No | 11.6 | 14.3 | 22.8 | No | 8.8 | 3 | 13.8 | | |
| Eye Color | | | | | | | | | 21.15 | 0.007 |
| Pearl | No | 7.2 | 14.3 | 11.4 | No | 2.9 | 3.5 | 7.2 | | |
| Brown | 25 | 4.4 | 33.3 | No | 33.3 | 38.2 | 29.1 | 11.6 | | |
| Orange | 29.5 | 40.6 | 28.6 | 42.9 | 23.8 | 14.7 | 27.9 | 34.8 | | |
| Red | 45.5 | 47.8 | 23.8 | 45.7 | 42.9 | 44.2 | 39.5 | 46.4 | | |
| Shank Color | | | | | | | | | 16.20 | 0.028 |
| Black | 20.5 | 22.4 | 23.8 | 31 | 14.3 | 25.2 | 19.8 | 36.2 | | |
| White | 15.9 | 26.8 | 19 | 16 | 23.8 | 26.5 | 18.6 | 23.6 | | |
| Grey | 9.1 | No | 9.6 | 2.9 | No | 4.2 | 6.9 | 0.2 | | |
| Yellow | 54.5 | 50.8 | 47.6 | 50.1 | 61.9 | 44.1 | 54.7 | 40 | | |
| Skin Color | | | | | | | | | 19.41 | 0.003 |
| White | 70.5 | 68.1 | 76.2 | 54.3 | 57.1 | 47.1 | 68.3 | 59.4 | | |
| Yellow | 28.2 | 26.1 | 23.8 | 40 | 40.9 | 52.9 | 30.4 | 35.5 | | |
| Pink | 1.3 | 5.8 | No | 5.7 | 2 | No | 1.3 | 5.1 | | |
| Feather Distribution | | | | | | | | | | |
| Normal | 100 | 100 | 100 | 100 | 100 | 100 | 100 | | | |
| Shank feather | | | | | | | | | 9.04 | 0.06 |

*(Continued)*

**Table 3.** (Continued)

| Agroecology | | | | | | | | | |
| --- | --- | --- | --- | --- | --- | --- | --- | --- | --- |
| Qualitative trait | Highland | | Midland | | Lowland | | Overall | | $\chi^2$ | p-value |
| Present | 27.3 | 40.6 | 33.3 | 57.1 | 47.6 | 58.8 | 33.7 | 49.3 | | |
| Absent | 72.7 | 59.4 | 66.7 | 42.9 | 52.4 | 41.2 | 66.3 | 50.7 | | |

No = not observed, $\chi^2$ = chi-square.

in Ethiopia supported by [45,51,84,85], highlighting the distinct morphological characteristics of indigenous chickens in Ethiopia. The observed variations in body weight, body dimensions, and sexual dimorphism reflect the genetic diversity and adaptability of these chicken populations to diverse agroecological conditions [84–87].

## Correlations of body weight and linear body measurements

The pearson correlation results presented in the Table 4 provide insights into the relationships between body weight (BW) and various linear body measurements of the study area The Pearson ea reveals several significant positive associations. The strongest correlations are observed between BW and Wither's Height (WS, r = 0.476, p < 0.01), Body Length (BL, r = 0.438, p < 0.01), and Body Depth (BD, r = 0.425, p < 0.01). These findings are consistent with a recent study by [66], who reported similar strong correlations between body weight and height, length, and depth measurements in chicken. Body weight has a significant positive correlation with all the linear body measurements, indicating that as the body weight increases, the corresponding body measurements also tend to increase [88,89].

**Table 4. Linear body measurements of male and female chicken populations (N = 224).**

| Morphometric traits | Sex | Agroecologies | | | | | |
| --- | --- | --- | --- | --- | --- | --- | --- |
| | | Highland nm = 44 nf = 68 | Midland nm = 44 nf = 68 | Lowland nm = 44 nf = 68 | Overall nm = 138 nf = 86 | F-v | P-values |
| Bodyweight(kg) | M | 1.54±0.04 | 1.43±0.04 | 1.33±0.02 | 1.46±0.02 | 5.449 | 0.006 |
| | F | 1.15±0.017 | 1.21±0.02 | 1.22±0.02 | 1.21±0.01 | 2.909 | 0.05 |
| Body lengths(cm) | M | 37.38±0.48 | 38.49±0.23 | 39.0±0.18 | 38.05±0.26 | 3.705 | 0.029 |
| | F | 32.65±0.42 | 31.11±0.68 | 29.85±0.68 | 31.55±0.33 | 6.502 | 0.002 |
| Body width(cm) | M | 21.88±0.22 | 21.90±0.29 | 23.14±0.61 | 22.20±0.20 | 3.566 | 0.033 |
| | F | 18.78±0.30 | 19.68±0.47 | 18.88±0.34 | 19.04±0.21 Ns | 1.609 | 0.204 |
| Shank length(cm) | M | 8.53±0.16 | 8.72±0.23 | 9.31±0.17 | 8.76±0.11 | 4.368 | 0.016 |
| | F | 6.6±0.08 | 6.81±0.11 | 6.64±0.11 | 6.66±0.05* | 1.211 | 0.301 |
| Shank. Circ | M | 3.57±0.08 | 3.66±0.104 | 3.52±0.09 | 3.58±0.05 | .39 | 0.672 |
| | F | 3.37±0.06 | 3.31±0.06 | 3.37±0.06 | 3.36±0.03 Ns | .273 | 0.762 |
| Wingspan(cm) | M | 38.65±0.21 | 38.51±0.25 | 38.43±0.23 | 38.56±0.13 | .223 | 0.801 |
| | F | 33.09±0.18 | 33.06±0.26 | 32.69±0.27 | 32.98±0.13 Ns | .809 | 0.448 |
| Comp length(cm) | M | 2.20±0 .04 | 2.55±0.08 | 2.36±0.07 | 2.32±0.040 | 7.665 | 0.001 |
| | F | 2.02±0.04 | 2.15±0.02 | 2.15±0.27 | 2.08±0.024 | 3.930 | 0.022 |
| Beak length(cm) | M | 1.85±0.03 | 1.96±0.04 | 1.99±0.03 | 1.91±0.023 | 3.660 | 0.030 |
| | F | 1.31±0.02 | 1.25±0.032 | 1.23±0.03 | 1.27±0.019 Ns | 1.527 | 0.221 |

M = male, F = female, nm = number of male, nf = number of female, F-V = F-values, ns = non significance.

*significance (p<0.05) and significance at (p<0.01).

**Table 5. Pearson correlations of body weight and linear body measurements of the study area.**

| Variable | BW | BL | BD | SHL | SHC | WS | CL | BKL |
|---|---|---|---|---|---|---|---|---|
| BW | 1 | .44 | .425 | .452 | .309 | .416 | .153 | .279 |
| BL | | 1 | .649 | .674 | .453 | .721 | .315 | .537 |
| BD | | | 1 | .697 | .290 | .617 | .235 | .443 |
| SHL | | | | 1 | .231 | .753 | .320 | .646 |
| SHC | | | | | 1 | .429 | .197 | .228 |
| WS | | | | | | 1 | .337 | .720 |
| CL | | | | | | | 1 | .300 |
| BKL | | | | | | | | 1 |

Correlation is significant at the 0.01 level (2-tailed). Correlation is significant at the 0.05 level (2-tailed). BW = Body Weight, BL = Body Length, BD = Body width; WS = Wing Span, CL = comp length ShL = Shank Length, ShC = Shank Circumference, BKL = beak length.

Body weight had the strongest correlations with shank length, body length, and wing span (Table 5), indicating these traits could be valuable indirect selection criteria to improve overall body size and productivity in indigenous chicken populations [90]. Moderate positive correlations were also observed between body weight and body width as well as shank circumference, further highlighting the interrelated nature of these morphological traits [91] (Table 4). The findings suggest that selecting for increased body weight could lead to correlated improvements in other desirable linear body measurements, which may enhance the market value and overall productivity of these chicken breeds [92].

Numerous studies have reported positive correlations between various body measurements in indigenous chicken populations, which aligns with the findings of the present study and [84] examined the phenotypic characterization of indigenous chicken ecotypes in the Central Highlands of Ethiopia and found significant positive correlations between body weight and other linear body traits, such as body length (r = 0.44, p < 0.01), body width (r = 0.43, p < 0.01), and wing span (r = 0.41, p < 0.01). Similarly, [85] reported a high positive correlation between body weight and shank length (r = 0.45, p < 0.01) in their study on indigenous chickens in the highlands of Ethiopia, consistent with the findings of (1) on other indigenous chicken populations in the country.

These correlations between body weight and linear body measurements have important implications for animal breeding and selection. As [93] highlighted, it is common practice in animal breeding to predict body weight from linear body measurements, as this can contribute to improvements in economically important traits like productivity. This is particularly relevant for indigenous chicken populations, as [94] emphasized the increasing focus on selecting for higher body weight to enhance the productivity of these important genetic resources [88].

## Conclusions and recommendations

### Conclusions

Based on the key findings presented, the following conclusions can be drawn:

The cluster analysis of indigenous chickens in the Liben Jawi district reveals significant insights into the phenotypic diversity and potential breeding strategies for local poultry and provides a foundation for future breeding programs.

Based on the specific traits identified through cluster and multivariate analysis of indigenous chickens, four distinct clusters were identified in the study district, categorized by body size and morphological indices.

The multivariate analyses of indigenous chickens revealed notable phenotypic diversity and relationships between morphological indices and their functional significance.

The findings underscore the importance of understanding the relationship between morphological traits.

The variations in body size, shape, and other characteristics not only reflect adaptive strategies to environmental challenges but also reveal the influences of local conditions and farmer preferences on breeding practices.

The study found that male chickens (cocks) were heavier than female chickens (hens), indicating the presence of clear sexual dimorphism in the indigenous chicken population.

The study revealed positive correlations between various linear body measurements, such as body length, body width, shank length, and wing span, suggesting changes in one parameter are likely to be reflected in the others, which is important for understanding the relationship between different phenotypic traits.

The study found significant positive correlations between body weight and other linear body measurements indicating that selecting for any of these parameters could lead to improvements in body weight, which is a crucial production trait.

The findings emphasize the significance of considering both qualitative (sexual dimorphism) and quantitative (body weight and linear measurements) aspects of chicken phenotypes. By recognizing and utilizing these phenotypic traits, farmers and breeders can make informed decisions to improve the genetic potential of indigenous chickens. Generally, the study provides valuable insights into the phenotypic characteristics of indigenous chickens in the Liban Jawi District, Ethiopia, which can contribute to the development of targeted breeding programs and the enhancement of local poultry productivity.

The socio-economic characteristics of smallholder chicken producers can vary significantly across agroecological zones, which may have important implications for the design and implementation of development interventions targeting chicken production systems.

These findings highlight the impact of local environmental conditions and farmer preferences on the phenotypic characteristics of indigenous chickens, emphasizing the intricate relationship between morphology, function, and adaptive strategies in poultry management.

## Recommendations

Based on the study of the phenotypic characterization of indigenous chickens in Liban Jawi District, Ethiopia, the following recommendations can be made. Leverage sexual dimorphism and correlations as positive correlations were observed between sexual dimorphism and body traits by prioritizing body weight as a key selection criterion due to its strong positive correlations to enhance the genetic potential of the indigenous chicken population.

Implement community-based breeding programs to disseminate research findings and provide training to farmers on effective breeding strategies that leverage the identified traits to support the development of targeted breeding programs.

Incorporate the insights gained from the study into national poultry development policies to improve the productivity and sustainability of the local industry for enhancing food security and livelihoods for smallholder farmers.

Capitalize on the strong correlations between chicken body weight and linear measurements to develop simple, low-cost assessment methods to empower farmers to monitor their flocks and make informed management and marketing decisions, addressing the need for education and training highlighted in the conclusions.

Design extension and training programs tailored to less-educated farmers to ensure equitable access to productivity-enhancing information to align with the emphasis on educating smallholder farmers about phenotypic traits.

Recognize the central role of women in smallholder chicken production by developing gender-responsive support services and technologies to maximize the benefits of chicken enterprises for household nutrition and income generation, reflecting the socio-economic aspects mentioned in the conclusions.

Integrate contextual factors such as agroecology and socioeconomics into extension and technology dissemination strategies to enhance the relevance and adoption of improved chicken management practices, translating research insights into practical actions that benefit the target population.

Undertake further research to explore the underlying genetic factors contributing to phenotypic variations in indigenous chickens to align with the need for a deeper understanding of genetic resources as indicated in the conclusions.

## Supporting information

**S1 Table. Traits of chickens by agro-ecological zones.**
(DOCX)

## Acknowledgments

The authors would like to extend their appreciation to the Mamo Mezemir Campus of Ambo University, Ethiopia for permitting to conduct the case as an MSc study.

## Author Contributions

**Conceptualization:** Desalegn Begna.

**Data curation:** Teferi Bacha.

**Formal analysis:** Teferi Bacha.

**Investigation:** Desalegn Begna.

**Methodology:** Desalegn Begna.

**Project administration:** Kasahun Bekana.

**Resources:** Shamble Boki.

**Supervision:** Desalegn Begna.

**Writing – original draft:** Desalegn Begna, Teferi Bacha.

**Writing – review & editing:** Desalegn Begna, Teferi Bacha, Shamble Boki.

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
