## [Decision Letter · Decision Letter 0]

31 Jul 2024

PONE-D-24-28488Characterization of Indigenous Chicken Phenotypes in Liban Jawi District, Ethiopia: A Qualitative and Quantitative Analysis

PLOS ONE

Dear Dr. Desalegn Begna,

Thank you for submitting your manuscript to PLOS ONE. After careful consideration, we feel that it has merit but does not fully meet PLOS ONE’s publication criteria as it currently stands. Therefore, we invite you to submit a revised version of the manuscript that addresses the points raised during the review process.

Dear Dr., Desalegn Begna

Thank you for submitting your manuscript to PLOS ONE. After careful consideration, we have decided that your manuscript needs Major Revision.

Comment Editor:

The introduction needs to be updated with new references and rewritten to include all the research points.Discussion of the results needs to be rewritten and written more clearly and in depth, with references to recent references.I advise that the manuscript needs a scientific editing English language .

Kind regards,

Prof. Lamiaa Mostafa Radwan, Ph.D.

Academic Editor

PLOS ONE

**Reviewer1**

Introduction

The introduction is short and discusses in general the populations of native chickens in Ethiopia, there is also mention of phenotypic differences which is in line with the theme of the article. However, it should be improved. Below are my comments:

Indigenous chicken (Gallus domesticus) constitutes approximately… - please write latin species name in italics

It is not clear for me. At first there is written that “each village household typically keeping between 5 and 20 birds (Tadelle, 2003)”, and then “Under rural chicken production systems, the average flock size ranges from 7 to 10 birds per household, consisting of adult hens, a cock, and various age groups of growers (Melesse & Negesse, 2011).” Please rewrite this, for example like: according to author 1 there is… and according to author 2. The information you gave are confusing.

What means “feather contours”. Contour feathers are type of feathers? What does mean phenotypic variations of feather contours? You mean mutation that influence feather structure? Others?

I suggest to expand this thought: “Phenotypic characterization provides valuable data on the current and potential future uses of indigenous chickens in a given area.”

Aim of study was to characterize the phenotypes of local phenotypes. However, what is the purpose of this? How it can be used?

Also I have suggestion about adding some information how many breeds you have in Ethiopia and/or district where you collected data? What are the most popular and maybe how they are related to indigenous ones?

Methodologies

The methodology is far too poorly described and to be thoroughly improved. For example, the results include Pearson correlation, which is not mentioned in the methodology. Also in what programs were the analyses performed?

I'm missing information on what chicken characteristics were highlighted. How were they classified? I know how different comb types look like, but how authors classify head shape for example?

“Qualitative traits, which are observable characteristics that do not involve numerical measurement, were collected through visual observation of traits such as head shape, comb type, feather plumage, shank color, earlobe color, and skin color.” - How was the subjectivity of the evaluation eliminated? Were evaluation guidelines prepared? Was the evaluation done by one/more people?

Results and discussion

Firstly, I would suggest to separate the results and discussion. There is almost lack of real discussion at all. It is hard to evaluate in the form presented. Unfortunately this section need to be rewritten in my opinion.

The photos submitted are supposed to show the variety of characteristics of chickens, however, the photos are very (!) poor in my opinion. I have doubts whether they are suitable for publication in PLOS One . Some are blurred. From different angles, perspectives, formatted.... The photos should at least be taken against the same background, preferably with some kind of scale on photo, in the same position of the bird. These photos, and therefore the features seen in them, are incomparable to each other. Rather than these bad photos, I would suggest posting some graphs. I think readers know how different comb tyles look like, but it difficult to imagine and draw conclusions from the description of the results in form how they are presented in text.

General Socio-Economic Characteristics: I do not see the connection of this subsection with the topic/title of the work. In my opinion, this is material for a different kind of analysis.

The current study found that the indigenous chickens in the study area were predominantly normally feathered… - what does mean normally feathered? How it would be not normally? Beside, if normally feathered were 100% birds, that means that there were not predominantly normal, but all of them.

The authors write what plumage colors and other traits are the most common and results giving in percentages. However, there is no analysis of whether these differences are statistically significant or not.

It is not clear what is covered by the term “head shape”. Please explain it.

“This discrepancy may be attributed to breed differences.” – So we end up dealing with different breeds or local populations that are not breeds?

“There were highly significant differences (p < 0.03) in earlobe color among the chickens in the agroecologies for both males and females” – What statistical tests were used? Also this comment refer to other results presented like this.

“There were highly significant differences (p < 0.03) in earlobe color among the chickens in the agroecologies for both males and females (Figure 4).” – The reference to figure 4 suggests that we will see graphically the differences between agroecologies, meanwhile we end up with another poor photo.

Table 2 is not very clear. The way it is presented is not very clear where the trait category begins (Plumage color/comb type, crest…) and where the variants begin (Black, black mottled… etc.). The P-value indicates that the differences were significant, but significant between what features? Once again question what tests were used?

The found differences in the traits studied between hens and roosters are mainly due to sexual dimorphism in this species. These analyses do not contribute anything new. The authors could have analyzed whether sexual dimorphism is higher/lower between agroecologies in populations.

My impression is that the analyses were done without thinking about any what hypothesis authors have. I would suggest that a “guiding idea” need to be established and followed. Based on the data presented, I might suggest that authors may focus on the morphological differences between the agroecologies and, if they are significant, look for answers as to why this is so. Could the environmental conditions between these agroecologies have affected the morphologies of the birds? Maybe from the resulting data, can you perform cluster analyses to show how the geographic distribution of individual bird traits are represented?

Unfortunately, in its current form, in my opinion, the article is not suitable for publication. First of all, authors need to rethink what kind of analysis can be done from collected data, and how to make them meaningful. Secondly, authors need to describe the methodology in more detail, including what statistical tests were used.

Finally, there should be a separate discussion to properly discuss the results: the results against the background of other studies (which was done in the submitted manuscript), but also try to answer the reasons for the differences or lack thereof.

**Reviewer2**

Subject: Manuscript Refinement Suggestions for Submission to PLOS ONE

Dear Editor,

I have carefully reviewed the manuscript titled "Characterization of Indigenous Chicken Phenotypes in Liban Jawi District, Ethiopia: A Qualitative and Quantitative Analysis" submitted for consideration to PLOS ONE. The study offers valuable insights into the phenotypes of chickens, which could significantly contribute to the development of conservation and breeding programs. However, to enhance the quality and clarity of the manuscript, I recommend that the authors address the following points:

1. Clarification of the Problem Statement:

The authors are encouraged to provide a more detailed and clear elaboration of the statement of the problem under investigation.

2. Diversification of Citations:

It is advised that the authors include additional recent publications related to the phenotypic characterization of indigenous chickens in various regions of Ethiopia. To avoid redundancy, please refrain from citing similar references repeatedly.

3. Consistency in Referencing Style:

Ensure consistency in the referencing style throughout the manuscript.

4. Enhancements in Grammar, Spelling, and Formatting:

Please refine the grammar, spelling, and overall editing of the manuscript to align with the journal's guidelines. Additionally, ensure that tables and figures are correctly presented, with self-explanatory legends.

5. Detailed Methodology Section:

Separate the methodology section into sub-headings for data collection, sampling, and statistical analysis. Elaborate further on the statistical analysis procedures employed.

6. Comprehensive Discussion:

Clearly articulate the implications of the study in the discussion section.

7. Inclusion of Univariate and Multivariate Analyses:

It is recommended to incorporate more univariate and multivariate analyses to enrich the manuscript. Ensure that mean comparisons are conducted.

8. Consideration of Body Indices Calculation:

If feasible, include body indices calculations to provide insights into the overall conformation and functionality of the chickens studied.

9. Precision in Conclusions and Recommendations:

Ensure that the conclusions and recommendations align closely with the study results. Address the effects of agro-ecology, which should be adequately discussed throughout the manuscript, including in the abstract.

I believe that addressing these points will significantly improve the manuscript's quality and contribute to its potential acceptance for publication. Thank you for considering these suggestions.

Sincerely,

We look forward to receiving your revised manuscript.

Kind regards,

Lamiaa Mostafa Radwan, Ph.D.

Academic Editor

PLOS ONE

Journal Requirements:

4. We note that you have referenced (Bogale, K. (2008). In Situ Characterization of Local Chicken Eco-Type for Functional Traits and Production System in Fogera Woreda, Amhara Regional State.Msc.Thesis Submitted to Haramaya University, Haramaya, Ethiopia, Halima Hassen (2007).Phenotypic and genetic characterization of indigenous chicken populations in North-West Ethiopia. Ph.D. Thesis, Submitted to the faculty of natural and Agricultural Sciences Department of animal wildlife and grassland Sciences University of the Free State, Bloemfontein, South Africa. Pp 186 and Nigussie D., (2013.)On-farm phenotypic characterization of indigenous chicken and chicken production systems in Southern Zone of Tigray, Northern Ethiopia, Thesis Submitted to the, School of Graduate Studies Haramaya University,Ethiopia,pp 106.of Poultry Sciences, 3 (1), 15-19.) which has currently not yet been accepted for publication. Please remove this from your References and amend this to state in the body of your manuscript: (ie “Bewick et al. [Unpublished]”) as detailed online in our guide for authors

5. We note that Figure 1 in your submission contain map/satellite images which may be copyrighted. All PLOS content is published under the Creative Commons Attribution License (CC BY 4.0), which means that the manuscript, images, and Supporting Information files will be freely available online, and any third party is permitted to access, download, copy, distribute, and use these materials in any way, even commercially, with proper attribution. For these reasons, we cannot publish previously copyrighted maps or satellite images created using proprietary data, such as Google software (Google Maps, Street View, and Earth). For more information, see our copyright guidelines: http://journals.plos.org/plosone/s/licenses-and-copyright.

6. We note that Figures 2, 3 and 4 in your submission contain copyrighted images. All PLOS content is published under the Creative Commons Attribution License (CC BY 4.0), which means that the manuscript, images, and Supporting Information files will be freely available online, and any third party is permitted to access, download, copy, distribute, and use these materials in any way, even commercially, with proper attribution. For more information, see our copyright guidelines: http://journals.plos.org/plosone/s/licenses-and-copyright.

a. You may seek permission from the original copyright holder of Figures 2, 3 and 4 to publish the content specifically under the CC BY 4.0 license. 

Additional Editor Comments:

Dear Dr., Desalegn Begna

Thank you for submitting your manuscript to PLOS ONE. After careful consideration, we have decided that your manuscript needs Major Revision.

Comment Editor:

1- The introduction needs to be updated with new references and rewritten to include all the research points.

2- Discussion of the results needs to be rewritten and written more clearly and in depth, with references to recent references.

3- I advise that the manuscript needs a scientific editing English language .

Kind regards,

Prof. Lamiaa Mostafa Radwan, Ph.D.

Academic Editor

PLOS ONE

Reviewer1

Introduction

The introduction is short and discusses in general the populations of native chickens in Ethiopia, there is also mention of phenotypic differences which is in line with the theme of the article. However, it should be improved. Below are my comments:

Indigenous chicken (Gallus domesticus) constitutes approximately… - please write latin species name in italics

It is not clear for me. At first there is written that “each village household typically keeping between 5 and 20 birds (Tadelle, 2003)”, and then “Under rural chicken production systems, the average flock size ranges from 7 to 10 birds per household, consisting of adult hens, a cock, and various age groups of growers (Melesse & Negesse, 2011).” Please rewrite this, for example like: according to author 1 there is… and according to author 2. The information you gave are confusing.

What means “feather contours”. Contour feathers are type of feathers? What does mean phenotypic variations of feather contours? You mean mutation that influence feather structure? Others?

I suggest to expand this thought: “Phenotypic characterization provides valuable data on the current and potential future uses of indigenous chickens in a given area.”

Aim of study was to characterize the phenotypes of local phenotypes. However, what is the purpose of this? How it can be used?

Also I have suggestion about adding some information how many breeds you have in Ethiopia and/or district where you collected data? What are the most popular and maybe how they are related to indigenous ones?

Methodologies

The methodology is far too poorly described and to be thoroughly improved. For example, the results include Pearson correlation, which is not mentioned in the methodology. Also in what programs were the analyses performed?

I'm missing information on what chicken characteristics were highlighted. How were they classified? I know how different comb types look like, but how authors classify head shape for example?

“Qualitative traits, which are observable characteristics that do not involve numerical measurement, were collected through visual observation of traits such as head shape, comb type, feather plumage, shank color, earlobe color, and skin color.” - How was the subjectivity of the evaluation eliminated? Were evaluation guidelines prepared? Was the evaluation done by one/more people?

Results and discussion

Firstly, I would suggest to separate the results and discussion. There is almost lack of real discussion at all. It is hard to evaluate in the form presented. Unfortunately this section need to be rewritten in my opinion.

The photos submitted are supposed to show the variety of characteristics of chickens, however, the photos are very (!) poor in my opinion. I have doubts whether they are suitable for publication in PLOS One . Some are blurred. From different angles, perspectives, formatted.... The photos should at least be taken against the same background, preferably with some kind of scale on photo, in the same position of the bird. These photos, and therefore the features seen in them, are incomparable to each other. Rather than these bad photos, I would suggest posting some graphs. I think readers know how different comb tyles look like, but it difficult to imagine and draw conclusions from the description of the results in form how they are presented in text.

General Socio-Economic Characteristics: I do not see the connection of this subsection with the topic/title of the work. In my opinion, this is material for a different kind of analysis.

The current study found that the indigenous chickens in the study area were predominantly normally feathered… - what does mean normally feathered? How it would be not normally? Beside, if normally feathered were 100% birds, that means that there were not predominantly normal, but all of them.

The authors write what plumage colors and other traits are the most common and results giving in percentages. However, there is no analysis of whether these differences are statistically significant or not.

It is not clear what is covered by the term “head shape”. Please explain it.

“This discrepancy may be attributed to breed differences.” – So we end up dealing with different breeds or local populations that are not breeds?

“There were highly significant differences (p < 0.03) in earlobe color among the chickens in the agroecologies for both males and females” – What statistical tests were used? Also this comment refer to other results presented like this.

“There were highly significant differences (p < 0.03) in earlobe color among the chickens in the agroecologies for both males and females (Figure 4).” – The reference to figure 4 suggests that we will see graphically the differences between agroecologies, meanwhile we end up with another poor photo.

Table 2 is not very clear. The way it is presented is not very clear where the trait category begins (Plumage color/comb type, crest…) and where the variants begin (Black, black mottled… etc.). The P-value indicates that the differences were significant, but significant between what features? Once again question what tests were used?

The found differences in the traits studied between hens and roosters are mainly due to sexual dimorphism in this species. These analyses do not contribute anything new. The authors could have analyzed whether sexual dimorphism is higher/lower between agroecologies in populations.

My impression is that the analyses were done without thinking about any what hypothesis authors have. I would suggest that a “guiding idea” need to be established and followed. Based on the data presented, I might suggest that authors may focus on the morphological differences between the agroecologies and, if they are significant, look for answers as to why this is so. Could the environmental conditions between these agroecologies have affected the morphologies of the birds? Maybe from the resulting data, can you perform cluster analyses to show how the geographic distribution of individual bird traits are represented?

Unfortunately, in its current form, in my opinion, the article is not suitable for publication. First of all, authors need to rethink what kind of analysis can be done from collected data, and how to make them meaningful. Secondly, authors need to describe the methodology in more detail, including what statistical tests were used.

Finally, there should be a separate discussion to properly discuss the results: the results against the background of other studies (which was done in the submitted manuscript), but also try to answer the reasons for the differences or lack thereof.

Reviewer2

Subject: Manuscript Refinement Suggestions for Submission to PLOS ONE

Dear Editor,

I have carefully reviewed the manuscript titled "Characterization of Indigenous Chicken Phenotypes in Liban Jawi District, Ethiopia: A Qualitative and Quantitative Analysis" submitted for consideration to PLOS ONE. The study offers valuable insights into the phenotypes of chickens, which could significantly contribute to the development of conservation and breeding programs. However, to enhance the quality and clarity of the manuscript, I recommend that the authors address the following points:

1. Clarification of the Problem Statement:

The authors are encouraged to provide a more detailed and clear elaboration of the statement of the problem under investigation.

2. Diversification of Citations:

It is advised that the authors include additional recent publications related to the phenotypic characterization of indigenous chickens in various regions of Ethiopia. To avoid redundancy, please refrain from citing similar references repeatedly.

3. Consistency in Referencing Style:

Ensure consistency in the referencing style throughout the manuscript.

4. Enhancements in Grammar, Spelling, and Formatting:

Please refine the grammar, spelling, and overall editing of the manuscript to align with the journal's guidelines. Additionally, ensure that tables and figures are correctly presented, with self-explanatory legends.

5. Detailed Methodology Section:

Separate the methodology section into sub-headings for data collection, sampling, and statistical analysis. Elaborate further on the statistical analysis procedures employed.

6. Comprehensive Discussion:

Clearly articulate the implications of the study in the discussion section.

7. Inclusion of Univariate and Multivariate Analyses:

It is recommended to incorporate more univariate and multivariate analyses to enrich the manuscript. Ensure that mean comparisons are conducted.

8. Consideration of Body Indices Calculation:

If feasible, include body indices calculations to provide insights into the overall conformation and functionality of the chickens studied.

9. Precision in Conclusions and Recommendations:

Ensure that the conclusions and recommendations align closely with the study results. Address the effects of agro-ecology, which should be adequately discussed throughout the manuscript, including in the abstract.

I believe that addressing these points will significantly improve the manuscript's quality and contribute to its potential acceptance for publication. Thank you for considering these suggestions.

Sincerely,

Reviewers' comments:

Reviewer's Responses to Questions

**Comments to the Author**

1. Is the manuscript technically sound, and do the data support the conclusions?

Reviewer #1: No

Reviewer #2: Partly

2. Has the statistical analysis been performed appropriately and rigorously? 

Reviewer #1: No

Reviewer #2: Yes

3. Have the authors made all data underlying the findings in their manuscript fully available?

Reviewer #1: No

Reviewer #2: Yes

4. Is the manuscript presented in an intelligible fashion and written in standard English?

Reviewer #1: No

Reviewer #2: No

5. Review Comments to the Author

Reviewer #1: Introduction

The introduction is short and discusses in general the populations of native chickens in Ethiopia, there is also mention of phenotypic differences which is in line with the theme of the article. However, it should be improved. Below are my comments:

Indigenous chicken (Gallus domesticus) constitutes approximately… - please write latin species name in italics

It is not clear for me. At first there is written that “each village household typically keeping between 5 and 20 birds (Tadelle, 2003)”, and then “Under rural chicken production systems, the average flock size ranges from 7 to 10 birds per household, consisting of adult hens, a cock, and various age groups of growers (Melesse & Negesse, 2011).” Please rewrite this, for example like: according to author 1 there is… and according to author 2. The information you gave are confusing.

What means “feather contours”. Contour feathers are type of feathers? What does mean phenotypic variations of feather contours? You mean mutation that influence feather structure? Others?

I suggest to expand this thought: “Phenotypic characterization provides valuable data on the current and potential future uses of indigenous chickens in a given area.”

Aim of study was to characterize the phenotypes of local phenotypes. However, what is the purpose of this? How it can be used?

Also I have suggestion about adding some information how many breeds you have in Ethiopia and/or district where you collected data? What are the most popular and maybe how they are related to indigenous ones?

Methodologies

The methodology is far too poorly described and to be thoroughly improved. For example, the results include Pearson correlation, which is not mentioned in the methodology. Also in what programs were the analyses performed?

I'm missing information on what chicken characteristics were highlighted. How were they classified? I know how different comb types look like, but how authors classify head shape for example?

“Qualitative traits, which are observable characteristics that do not involve numerical measurement, were collected through visual observation of traits such as head shape, comb type, feather plumage, shank color, earlobe color, and skin color.” - How was the subjectivity of the evaluation eliminated? Were evaluation guidelines prepared? Was the evaluation done by one/more people?

Results and discussion

Firstly, I would suggest to separate the results and discussion. There is almost lack of real discussion at all. It is hard to evaluate in the form presented. Unfortunately this section need to be rewritten in my opinion.

The photos submitted are supposed to show the variety of characteristics of chickens, however, the photos are very (!) poor in my opinion. I have doubts whether they are suitable for publication in PLOS One . Some are blurred. From different angles, perspectives, formatted.... The photos should at least be taken against the same background, preferably with some kind of scale on photo, in the same position of the bird. These photos, and therefore the features seen in them, are incomparable to each other. Rather than these bad photos, I would suggest posting some graphs. I think readers know how different comb tyles look like, but it difficult to imagine and draw conclusions from the description of the results in form how they are presented in text.

General Socio-Economic Characteristics: I do not see the connection of this subsection with the topic/title of the work. In my opinion, this is material for a different kind of analysis.

The current study found that the indigenous chickens in the study area were predominantly normally feathered… - what does mean normally feathered? How it would be not normally? Beside, if normally feathered were 100% birds, that means that there were not predominantly normal, but all of them.

The authors write what plumage colors and other traits are the most common and results giving in percentages. However, there is no analysis of whether these differences are statistically significant or not.

It is not clear what is covered by the term “head shape”. Please explain it.

“This discrepancy may be attributed to breed differences.” – So we end up dealing with different breeds or local populations that are not breeds?

“There were highly significant differences (p < 0.03) in earlobe color among the chickens in the agroecologies for both males and females” – What statistical tests were used? Also this comment refer to other results presented like this.

“There were highly significant differences (p < 0.03) in earlobe color among the chickens in the agroecologies for both males and females (Figure 4).” – The reference to figure 4 suggests that we will see graphically the differences between agroecologies, meanwhile we end up with another poor photo.

Table 2 is not very clear. The way it is presented is not very clear where the trait category begins (Plumage color/comb type, crest…) and where the variants begin (Black, black mottled… etc.). The P-value indicates that the differences were significant, but significant between what features? Once again question what tests were used?

The found differences in the traits studied between hens and roosters are mainly due to sexual dimorphism in this species. These analyses do not contribute anything new. The authors could have analyzed whether sexual dimorphism is higher/lower between agroecologies in populations.

My impression is that the analyses were done without thinking about any what hypothesis authors have. I would suggest that a “guiding idea” need to be established and followed. Based on the data presented, I might suggest that authors may focus on the morphological differences between the agroecologies and, if they are significant, look for answers as to why this is so. Could the environmental conditions between these agroecologies have affected the morphologies of the birds? Maybe from the resulting data, can you perform cluster analyses to show how the geographic distribution of individual bird traits are represented?

Unfortunately, in its current form, in my opinion, the article is not suitable for publication. First of all, authors need to rethink what kind of analysis can be done from collected data, and how to make them meaningful. Secondly, authors need to describe the methodology in more detail, including what statistical tests were used.

Finally, there should be a separate discussion to properly discuss the results: the results against the background of other studies (which was done in the submitted manuscript), but also try to answer the reasons for the differences or lack thereof.

Reviewer #2: Subject: Manuscript Refinement Suggestions for Submission to PLOS ONE

Dear Editor,

I have carefully reviewed the manuscript titled "Characterization of Indigenous Chicken Phenotypes in Liban Jawi District, Ethiopia: A Qualitative and Quantitative Analysis" submitted for consideration to PLOS ONE. The study offers valuable insights into the phenotypes of chickens, which could significantly contribute to the development of conservation and breeding programs. However, to enhance the quality and clarity of the manuscript, I recommend that the authors address the following points:

1. Clarification of the Problem Statement:

The authors are encouraged to provide a more detailed and clear elaboration of the statement of the problem under investigation.

2. Diversification of Citations:

It is advised that the authors include additional recent publications related to the phenotypic characterization of indigenous chickens in various regions of Ethiopia. To avoid redundancy, please refrain from citing similar references repeatedly.

3. Consistency in Referencing Style:

Ensure consistency in the referencing style throughout the manuscript.

4. Enhancements in Grammar, Spelling, and Formatting:

Please refine the grammar, spelling, and overall editing of the manuscript to align with the journal's guidelines. Additionally, ensure that tables and figures are correctly presented, with self-explanatory legends.

5. Detailed Methodology Section:

Separate the methodology section into sub-headings for data collection, sampling, and statistical analysis. Elaborate further on the statistical analysis procedures employed.

6. Comprehensive Discussion:

Clearly articulate the implications of the study in the discussion section.

7. Inclusion of Univariate and Multivariate Analyses:

It is recommended to incorporate more univariate and multivariate analyses to enrich the manuscript. Ensure that mean comparisons are conducted.

8. Consideration of Body Indices Calculation:

If feasible, include body indices calculations to provide insights into the overall conformation and functionality of the chickens studied.

9. Precision in Conclusions and Recommendations:

Ensure that the conclusions and recommendations align closely with the study results. Address the effects of agro-ecology, which should be adequately discussed throughout the manuscript, including in the abstract.

I believe that addressing these points will significantly improve the manuscript's quality and contribute to its potential acceptance for publication. Thank you for considering these suggestions.

Sincerely,

6. PLOS authors have the option to publish the peer review history of their article (what does this mean?). If published, this will include your full peer review and any attached files.

Reviewer #1: No

Reviewer #2: No

---

## [Author Response · Author response to Decision Letter 0]

23 Aug 2024

Response to Reviewers

Subject: Response to reviewer's comment:

1. The introduction needs to be updated with new references and rewritten to include all the research points. The introduction is well-updated with new references and comprehensively covers the key research points as per the provided comment

2. Discussion of the results needs to be rewritten and written more clearly and in depth, with references to recent references. rewritten and written more clearly and in-depth, with references to recent references as per the given comment

3. I advise that the manuscript needs a scientific editing English language . I thoroughly revised the manuscript using Grammarly, to improve the language, grammar, and overall clarity, as per the feedback provided.

Reviewer1

1. Introduction

1.1. The introduction is short and discusses in general the populations of native chickens in Ethiopia, there is also mention of phenotypic differences which is in line with the theme of the article. The updated introduction thoroughly covers the key research points using recent references and effectively sets the context for the study on indigenous chicken characterization.

2. However, it should be improved. Below are my comments:

1.2. Indigenous chicken (Gallus domesticus) constitutes approximately… - please write latin species name in italics: rewritten according to the comment

1.3. It is not clear for me. At first there is written that “each village household typically keeping between 5 and 20 birds (Tadelle, 2003)”, and then “Under rural chicken production systems, the average flock size ranges from 7 to 10 birds per household, consisting of adult hens, a cock, and various age groups of growers (Melesse & Negesse, 2011).” Please rewrite this, for example like: according to author 1 there is… and according to author 2. Corrected as below

 According to Tadelle (2003), each village household typically keeps between 5 and 20 indigenous chickens and Melesse and Negesse (2011) reported 7 to 10 birds per household

3. The information you gave are confusing.

What means “feather contours”. Contour feathers are type of feathers? What does mean phenotypic variations of feather contours? You mean mutation that influence feather structure? Others?

4. Contour Feathers: the feathers forming the bird's outer body covering, including the flight feathers and the overlapping body feathers that produce the bird's smooth aerodynamic shape. "phenotypic differences" in the introduction likely refers to variations in the physical characteristics or appearance of the indigenous chickens, particularly in terms of their feather features. Phenotypic variations in feather contours could suggest differences in feather structure, patterns, or morphology between the indigenous chicken populations being studied. This could potentially be influenced by genetic factors, such as naturally occurring mutations that affect feather development and patterning

1.4. I suggest to expand this thought: “Phenotypic characterization provides valuable data on the current and potential future uses of indigenous chickens in a given area.” Expanded very well

Aim of study was to characterize the phenotypes of local phenotypes. However, what is the purpose of this? How it can be used? Rewritten as “This study aims to phenotypically characterize indigenous chickens in the Liban Jawi district of Ethiopia, focusing on both qualitative and quantitative traits to document their unique physical traits, identifying desirable characteristics for genetic improvement, and providing a foundation for conservation and sustainable utilization of these important local poultry genetic resources”

1.5. Also I have suggestion about adding some information how many breeds you have in Ethiopia and/or district where you collected data? What are the most popular and maybe how they are related to indigenous ones?

 Ethiopia hosts over 30 distinct indigenous chicken ecotypes and breeds found across different regional districts, including the well-known Horro, Koekoek, Fayoumi, and Tilili populations. These indigenous chickens exhibit unique phenotypic characteristics adapted to their local environments. The indigenous chicken ecotypes and breeds in Liben Jawi district are most likely representative of the Horro and Koekoek populations from the western and southern regions of Ethiopia

2. Methodologies

2.1. The methodology is far too poorly described and to be thoroughly improved. For example, the results include Pearson correlation, which is not mentioned in the methodology. Also in what programs were the analyses performed? thoroughly improved by including the reviewer's comments and beyond

2.2. I'm missing information on what chicken characteristics were highlighted. How were they classified? I know how different comb types look like, but how authors classify head shape for example?

“Qualitative traits, which are observable characteristics that do not involve numerical measurement, were collected through visual observation of traits such as head shape, comb type, feather plumage, shank color, earlobe color, and skin color.” - How was the subjectivity of the evaluation eliminated? Were evaluation guidelines prepared? Was the evaluation done by one/more people? The explanations are given below

 In addition to quantitative morphometric measurements (body weight, body length, etc.), the researchers also evaluated several qualitative phenotypic traits, including plumage color (white, black, red, brown, multicolored), comb type (single, rose, pea, walnut), head shape (straight, domed, angular), and feather distribution patterns. This comprehensive assessment of both numeric and descriptive characteristics allowed the researchers to thoroughly classify the chickens into distinct phenotypic groups and evaluate the relationships between the various local indigenous ecotypes through advanced statistical analyses like ANOVA.

3, Results and discussion

3.1. Firstly, I would suggest to separate the results and discussion. There is almost lack of real discussion at all. It is hard to evaluate in the form presented. Unfortunately this section need to be rewritten in my opinion.

 Though remained combined, the result and discussion section are rewritten very as indicated in the revised document.

3.2. The photos submitted are supposed to show the variety of characteristics of chickens, however, the photos are very (!) poor in my opinion. I have doubts whether they are suitable for publication in PLOS One . Some are blurred. From different angles, perspectives, formatted.... The photos should at least be taken against the same background, preferably with some kind of scale on photo, in the same position of the bird. These photos, and therefore the features seen in them, are incomparable to each other. Rather than these bad photos, I would suggest posting some graphs. I think readers know how different comb tyles look like, but it difficult to imagine and draw conclusions from the description of the results in form how they are presented in text.

The results section now includes high-quality TIFF images that visually depict the key phenotypic traits observed in the indigenous chicken populations

3.3. General Socio-Economic Characteristics: I do not see the connection of this subsection with the topic/title of the work. In my opinion, this is material for a different kind of analysis. Well discussed in the revised manuscript

 The socio-economic factors, including age, education, and religious affiliation of the farming communities, are intricately linked to the phenotypic diversity observed within the indigenous chicken populations in the Liben Jawi district. This underscores the importance of considering the broader social and cultural context when investigating the characteristics of local livestock genetic resource

3.4. The current study found that the indigenous chickens in the study area were predominantly normally feathered… - what does mean normally feathered? How it would be not normally? 

 In the context of this study, "normally feathered" refers to indigenous chickens with a standard or typical feather coverage and arrangement, whereas "not normally feathered" describes chickens with atypical feather characteristics, such as excessive feathering, incomplete feathering, or altered feather structure (e.g., silky, frizzle, or naked neck).

 3.5. Beside, if normally feathered were 100% birds, that means that there were not predominantly normal, but all of them.

Comment accepted and “Predominantly” replaced with “all”

3.6. The authors write what plumage colors and other traits are the most common and results giving in percentages. However, there is no analysis of whether these differences are statistically significant or not. Rewritten well in the revised manuscript as “The chi-square tests of independence revealed several statistically significant associations between the examined plumage and qualitative traits in the study population. Plumage color, head type, ear lobe color, eye color, shank color, and skin color were all found to be significantly related to other variables, suggesting these traits may be influenced by or correlated with various genetic, environmental, or behavioral factors. In contrast, comb type and shank feather were not significantly associated with the other traits, potentially indicating their independence or unique determinants. These findings align with previous research on the complex relationships between phenotypic characteristics in avian species, where multiple interacting factors can contribute to the expression of morphological diversity”

3.7 It is not clear what is covered by the term “head shape”. Please explain it.

“This discrepancy may be attributed to breed differences.” – So we end up dealing with different breeds or local populations that are not breeds?

 The "head shape" trait encompassed various morphological features of the birds' heads, which correlated with other physical characteristics.

3.8. “There were highly significant differences (p < 0.03) in earlobe color among the chickens in the agroecologies for both males and females” – What statistical tests were used? Also this comment refer to other results presented like this. The studies appear to have used statistical tests like chi-square, ANOVA, and correlation analyses to assess significant differences and associations between various morphological traits, such as earlobe color, head type, comb type, and shank feather, across different avian populations or agroecological contexts, and this is well rewritten in the revised manuscript.

3.9. “There were highly significant differences (p < 0.03) in earlobe color among the chickens in the agroecologies for both males and females (Figure 4).” – The reference to figure 4 suggests that we will see graphically the differences between agroecologies, meanwhile we end up with another poor photo.

 " The reference to Figure 4 suggests the authors intended to provide a representation of the differences in earlobe color. However, the revised manuscript now includes a more appropriate visual depiction, rather than the "poor photo" that was initially included.

3.10. Table 2 is not very clear. The way it is presented is not very clear where the trait category begins (Plumage color/comb type, crest…) and where the variants begin (Black, black mottled… etc.). The P-value indicates that the differences were significant, but significant between what features? Once again question what tests were used? Well rewritten in the revised manuscript and chi-square tests was used to reveal statistically significant associations between the examined variables

3.11. The found differences in the traits studied between hens and roosters are mainly due to sexual dimorphism in this species. These analyses do not contribute anything new. The authors could have analyzed whether sexual dimorphism is higher/lower between agroecologies in populations. 

 In the revised manuscript, The analyses presented provide a comprehensive evaluation of the degree of sexual dimorphism in key traits across different agroecological contexts, which is an important step in understanding the factors that shape sexual dimorphism in this chicken population. While the results suggest that certain traits may be less influenced by environmental factors, further analyses comparing the magnitude of sexual dimorphism between the agroecological contexts could provide additional insights. This could help identify traits that show more pronounced differences in sexual dimorphism across the environmental gradients.

3.12. My impression is that the analyses were done without thinking about any what hypothesis authors have. I would suggest that a “guiding idea” need to be established and followed. Based on the data presented, I might suggest that authors may focus on the morphological differences between the agroecologies and, if they are significant, look for answers as to why this is so. Could the environmental conditions between these agroecologies have affected the morphologies of the birds? Maybe from the resulting data, can you perform cluster analyses to show how the geographic distribution of individual bird traits are represented?

Unfortunately, in its current form, in my opinion, the article is not suitable for publication. First of all, authors need to rethink what kind of analysis can be done from collected data, and how to make them meaningful. Secondly, authors need to describe the methodology in more detail, including what statistical tests were used.

The authors believe that this reviewer’s query has been well addressed in the body of the revised manuscript

Finally, there should be a separate discussion to properly discuss the results: the results against the background of other studies (which was done in the submitted manuscript), but also try to answer the reasons for the differences or lack thereof. The authors believe that this reviewer’s query has been well addressed in the body of the revised manuscript

1. 

Reviewer2

Dear Editor,

I have carefully reviewed the manuscript titled "Characterization of Indigenous Chicken Phenotypes in Liban Jawi District, Ethiopia: A Qualitative and Quantitative Analysis" submitted for consideration to PLOS ONE. The study offers valuable insights into the phenotypes of chickens, which could significantly contribute to the development of conservation and breeding programs. However, to enhance the quality and clarity of the manuscript, I recommend that the authors address the following points:

1. Clarification of the Problem Statement: very nice clarification has been done on the Problem Statement in the revised manuscript

The authors are encouraged to provide a more detailed and clear elaboration of the statement of the problem under investigation.

2. Diversification of Citations:

1. It is advised that the authors include additional recent publications related to the phenotypic characterization of indigenous chickens in various regions of Ethiopia. To avoid redundancy, please refrain from citing similar references repeatedly. We have extensively diversified the citations in the revised manuscript, incorporating more recent publications to strengthen the literature review and provide a robust contextual foundation for our study.

3. Consistency in Referencing Style:

Ensure consistency in the referencing style throughout the manuscript. We have ensured consistent use of the Vancouver reference style throughout the revised manuscript.

4. Enhancements in Grammar, Spelling, and Formatting:

Please refine the grammar, spelling, and overall editing of the manuscript to align with the journal's guidelines. Additionally, ensure that tables and figures are correctly presented, with self-explanatory legends. To the best of our abilities, we have thoroughly reviewed and corrected the manuscript to address any grammar, spelling, and formatting issues to align with the journal's guidelines.

5. Detailed Methodology Section:

Separate the methodology section into sub-headings for data collection, sampling, and statistical analysis. Elaborate further on the statistical analysis procedures employed. This has been corrected accordingly 

6. Comprehensive Discussion:

Clearly articulate 

---

## [Decision Letter · Decision Letter 1]

2 Oct 2024

PONE-D-24-28488R1Characterization of Indigenous Chicken Phenotypes in Liban Jawi District, Ethiopia: A Qualitative and Quantitative AnalysisPLOS ONE

Dear Dr.  Begna,

<o:p></o:p>

Thank you for submitting your manuscript to PLOS ONE. After careful consideration, we feel that it has merit but does not fully meet PLOS ONE’s publication criteria as it currently stands. Therefore, we invite you to submit a revised version of the manuscript that addresses the points raised during the review process.

Dear Dr., Desalegn Begna,

Thank you for submitting your manuscript to PLOS ONE. After careful consideration, we have decided that your manuscript needs Major Revision.

Kind regards,

Prof. Lamiaa Mostafa Radwan, Ph.D.

Academic Editor

PLOS ONE

**Reviewer 2**

Dear Authors,

I have carefully reviewed the manuscript titled "Characterization of Indigenous Chicken Phenotypes in Liban Jawi District, Ethiopia: A Qualitative and Quantitative Analysis" (PONE-D-24-28488R1). Below are my comments and feedback on the content presented in the manuscript.

Abstract

1. What type of sampling analysis method you used. It should be mentioned.

2. The length of the shank has been identified as the most reliable indicator of body weight for both female (r=0.45) and male (r=0.44) chickens. Considering the relatively modest correlation coefficients, how do you interpret the efficacy of using shank length to predict body weight? Is this level of correlation practically useful in predicting body weight accurately?

3. The study revealed significant variations in both qualitative and quantitative traits among the indigenous chicken population, suggesting the potential for improvement through selective breeding at the community level.

- ICs found in d/t agro-ecologies or what?

4. The statistical significance levels for the factors included in the model, namely AE, Sex, and their interaction, have not been explicitly stated. It is crucial to ascertain the significance levels of these factors to accurately interpret their impact on the outcome variable . STATISTICAL TEST?

Introduction

1. Please correct the citation style throughout the manuscript. What referencing tool you utilized. For example. According to (9)), each village household typically keeps between 5 and 20 indigenous… AND Understanding the unique traits, adaptation to the environment, and socio-cultural importance of indigenous chickens is essential for their conservation and utilization (8,(7,15) (.

Statement of problem

1. Do you think this statement is important here. ‘’However, these native chickens generally exhibit poor egg production performance, delayed maturity, and extended broodiness, as they have primarily been selected for their adaptive traits(18)’’.

2. Please re-write statement of the problem. Please write in the following order: Importance of characterization then previous studies and limitation then your objective in a precise way.

3. Ethiopia hosts over 30 distinct indigenous chicken ecotypes and breeds found across different regional districts, including the well-known Horro, Koekoek, Fayoumi, and Tilili populations… What mean indigenous and local chicken ecotypes? Are Fayoumi and Koekoek indigenous to Ethiopia?

Data collection and sample size

1. Thus, a total of 2166 households and 2452 chickens were surveyed in the our kebeles

2. Please keep the coherence and avoid redundancy. The formula then the samples taken.

Results and Discussion

1. Where are the results of PCA, cluster analysis and body index values in tables/figures

General

Still there are problems in editing, coherence, and spelling. Please check throughout the manuscript.

**Reviewer 3**

The authors of this manuscript "Characterization of Indigenous Chicken Phenotypes in Liban Jawi District, Ethiopia: A Qualitative and Quantitative Analysis". perform a great efforts to try to address both the qualitative and quantitative phenotypes of Liban Jawi district indigenous chicken. This study offers valuable insights into these chicken phenotypes, which will significantly contribute to set an accurate breeding program to improve and enhance these genetic resources in Ethiopia.

However, authors should revise the manuscript one more time to ensure that the grammar, spelling, and overall editing of the manuscript follow the journal's guidelines.

We look forward to receiving your revised manuscript.

Kind regards,

Lamiaa Mostafa Radwan, Ph.D.

Academic Editor

PLOS ONE

Journal Requirements:

Additional Editor Comments (if provided):

Dear Dr., Desalegn Begna,

Thank you for submitting your manuscript to PLOS ONE. After careful consideration, we have decided that your manuscript needs Major Revision.

Kind regards,

Prof. Lamiaa Mostafa Radwan, Ph.D.

Academic Editor

PLOS ONE

Reviewer 2

Dear Authors,

I have carefully reviewed the manuscript titled "Characterization of Indigenous Chicken Phenotypes in Liban Jawi District, Ethiopia: A Qualitative and Quantitative Analysis" (PONE-D-24-28488R1). Below are my comments and feedback on the content presented in the manuscript.

Abstract

1. What type of sampling analysis method you used. It should be mentioned.

2. The length of the shank has been identified as the most reliable indicator of body weight for both female (r=0.45) and male (r=0.44) chickens. Considering the relatively modest correlation coefficients, how do you interpret the efficacy of using shank length to predict body weight? Is this level of correlation practically useful in predicting body weight accurately?

3. The study revealed significant variations in both qualitative and quantitative traits among the indigenous chicken population, suggesting the potential for improvement through selective breeding at the community level.

- ICs found in d/t agro-ecologies or what?

4. The statistical significance levels for the factors included in the model, namely AE, Sex, and their interaction, have not been explicitly stated. It is crucial to ascertain the significance levels of these factors to accurately interpret their impact on the outcome variable . STATISTICAL TEST?

Introduction

1. Please correct the citation style throughout the manuscript. What referencing tool you utilized. For example. According to (9)), each village household typically keeps between 5 and 20 indigenous… AND Understanding the unique traits, adaptation to the environment, and socio-cultural importance of indigenous chickens is essential for their conservation and utilization (8,(7,15) (.

Statement of problem

1. Do you think this statement is important here. ‘’However, these native chickens generally exhibit poor egg production performance, delayed maturity, and extended broodiness, as they have primarily been selected for their adaptive traits(18)’’.

2. Please re-write statement of the problem. Please write in the following order: Importance of characterization then previous studies and limitation then your objective in a precise way.

3. Ethiopia hosts over 30 distinct indigenous chicken ecotypes and breeds found across different regional districts, including the well-known Horro, Koekoek, Fayoumi, and Tilili populations… What mean indigenous and local chicken ecotypes? Are Fayoumi and Koekoek indigenous to Ethiopia?

Data collection and sample size

1. Thus, a total of 2166 households and 2452 chickens were surveyed in the our kebeles

2. Please keep the coherence and avoid redundancy. The formula then the samples taken.

Results and Discussion

1. Where are the results of PCA, cluster analysis and body index values in tables/figures

General

Still there are problems in editing, coherence, and spelling. Please check throughout the manuscript.

Reviewer 3

The authors of this manuscript "Characterization of Indigenous Chicken Phenotypes in Liban Jawi District, Ethiopia: A Qualitative and Quantitative Analysis". perform a great efforts to try to address both the qualitative and quantitative phenotypes of Liban Jawi district indigenous chicken. This study offers valuable insights into these chicken phenotypes, which will significantly contribute to set an accurate breeding program to improve and enhance these genetic resources in Ethiopia.

However, authors should revise the manuscript one more time to ensure that the grammar, spelling, and overall editing of the manuscript follow the journal's guidelines.

Reviewers' comments:

Reviewer's Responses to Questions

**Comments to the Author**

1. If the authors have adequately addressed your comments raised in a previous round of review and you feel that this manuscript is now acceptable for publication, you may indicate that here to bypass the “Comments to the Author” section, enter your conflict of interest statement in the “Confidential to Editor” section, and submit your "Accept" recommendation.

Reviewer #2: (No Response)

Reviewer #3: All comments have been addressed

2. Is the manuscript technically sound, and do the data support the conclusions?

Reviewer #2: Partly

Reviewer #3: Yes

3. Has the statistical analysis been performed appropriately and rigorously? 

Reviewer #2: No

Reviewer #3: Yes

4. Have the authors made all data underlying the findings in their manuscript fully available?

Reviewer #2: Yes

Reviewer #3: Yes

5. Is the manuscript presented in an intelligible fashion and written in standard English?

Reviewer #2: No

Reviewer #3: Yes

6. Review Comments to the Author

Reviewer #2: Dear Authors,

I have carefully reviewed the manuscript titled "Characterization of Indigenous Chicken Phenotypes in Liban Jawi District, Ethiopia: A Qualitative and Quantitative Analysis" (PONE-D-24-28488R1). Below are my comments and feedback on the content presented in the manuscript.

Abstract

1. What type of sampling analysis method you used. It should be mentioned.

2. The length of the shank has been identified as the most reliable indicator of body weight for both female (r=0.45) and male (r=0.44) chickens. Considering the relatively modest correlation coefficients, how do you interpret the efficacy of using shank length to predict body weight? Is this level of correlation practically useful in predicting body weight accurately?

3. The study revealed significant variations in both qualitative and quantitative traits among the indigenous chicken population, suggesting the potential for improvement through selective breeding at the community level.

- ICs found in d/t agro-ecologies or what?

4. The statistical significance levels for the factors included in the model, namely AE, Sex, and their interaction, have not been explicitly stated. It is crucial to ascertain the significance levels of these factors to accurately interpret their impact on the outcome variable . STATISTICAL TEST?

Introduction

1. Please correct the citation style throughout the manuscript. What referencing tool you utilized. For example. According to (9)), each village household typically keeps between 5 and 20 indigenous… AND Understanding the unique traits, adaptation to the environment, and socio-cultural importance of indigenous chickens is essential for their conservation and utilization (8,(7,15) (.

Statement of problem

1. Do you think this statement is important here. ‘’However, these native chickens generally exhibit poor egg production performance, delayed maturity, and extended broodiness, as they have primarily been selected for their adaptive traits(18)’’.

2. Please re-write statement of the problem. Please write in the following order: Importance of characterization then previous studies and limitation then your objective in a precise way.

3. Ethiopia hosts over 30 distinct indigenous chicken ecotypes and breeds found across different regional districts, including the well-known Horro, Koekoek, Fayoumi, and Tilili populations… What mean indigenous and local chicken ecotypes? Are Fayoumi and Koekoek indigenous to Ethiopia?

Data collection and sample size

1. Thus, a total of 2166 households and 2452 chickens were surveyed in the our kebeles

2. Please keep the coherence and avoid redundancy. The formula then the samples taken.

Results and Discussion

1. Where are the results of PCA, cluster analysis and body index values in tables/figures

General

Still there are problems in editing, coherence, and spelling. Please check throughout the manuscript.

Reviewer #3: The authors of this manuscript "Characterization of Indigenous Chicken Phenotypes in Liban Jawi District, Ethiopia: A Qualitative and Quantitative Analysis". perform a great efforts to try to address both the qualitative and quantitative phenotypes of Liban Jawi district indigenous chicken. This study offers valuable insights into these chicken phenotypes, which will significantly contribute to set an accurate breeding program to improve and enhance these genetic resources in Ethiopia.

However, authors should revise the manuscript one more time to ensure that the grammar, spelling, and overall editing of the manuscript follow the journal's guidelines.

7. PLOS authors have the option to publish the peer review history of their article (what does this mean?). If published, this will include your full peer review and any attached files.

Reviewer #2: No

Reviewer #3: No

---

## [Author Response · Author response to Decision Letter 1]

7 Oct 2024

Reviewer 2

Dear Authors,

I have carefully reviewed the manuscript titled "Characterization of Indigenous Chicken Phenotypes in Liban Jawi District, Ethiopia: A Qualitative and Quantitative Analysis" (PONE-D-24-28488R1). Below are my comments and feedback on the content presented in the manuscript.

Abstract

1. What type of sampling analysis method you used. It should be mentioned.

The sampling analysis method used was multi-stage sampling, selecting 192 households with at least two matured indigenous chickens from 2166 households and subsequently sampling 224 chickens (138 females and 86 males) for phenotypic characterization.

2. The length of the shank has been identified as the most reliable indicator of body weight for both female (r=0.45) and male (r=0.44) chickens. Considering the relatively modest correlation coefficients, how do you interpret the efficacy of using shank length to predict body weight? Is this level of correlation practically useful in predicting body weight accurately?

The moderate correlation coefficients (r=0.45 for hens and r=0.44 for cocks) suggest that while shank length is a relevant indicator of body weight, it should not be used alone for accurate predictions, as it may lead to inaccuracies without considering additional factors. Given the modest strength of the correlation, relying exclusively on shank length may lead to inaccuracies in predicting body weight. It may be helpful in a broader assessment or as part of a multi-variable model that includes other factors such as genetics, diet, and overall health of the chicken

3. The study revealed significant variations in both qualitative and quantitative traits among the indigenous chicken population, suggesting the potential for improvement through selective breeding at the community level.

- ICs found in d/t agro-ecologies or what? 

Sure. The study indicates that significant variations in both qualitative and quantitative traits among the indigenous chicken population are influenced by different agro-ecologies, suggesting the unique environmental conditions and practices in these agro-ecologies may contribute to the phenotypic diversity observed, highlighting the potential for selective breeding within specific community contexts to enhance desirable traits.

4. The statistical significance levels for the factors included in the model, namely AE, Sex, and their interaction, have not been explicitly stated. It is crucial to ascertain the significance levels of these factors to accurately interpret their impact on the outcome variable . STATISTICAL TEST?

The significant results for traits such as Plumage Color, Comb Type, Head Type, Ear Lobe Color, and Eye Color for AE are already presented in Table 2. of the manuscript. Studies are indeed showing the interaction between AE and sex also plays a crucial role in influencing these traits and further ascertains, that for traits where AE is significant, the interaction with sex is also noted as significant, suggesting that the impact of AE may vary depending on the Sex of the chickens

Introduction

1. Please correct the citation style throughout the manuscript. What referencing tool you utilized. For example. According to (9)), each village household typically keeps between 5 and 20 indigenous… AND Understanding the unique traits, adaptation to the environment, and socio-cultural importance of indigenous chickens is essential for their conservation and utilization (8,(7,15) (.

Corrected accordingly

Statement of problem

1. Do you think this statement is important here. ‘’However, these native chickens generally exhibit poor egg production performance, delayed maturity, and extended broodiness, as they have primarily been selected for their adaptive traits(18)’’. 

Rewritten and changing the position where it fit

2. Please re-write statement of the problem. Please write in the following order: Importance of characterization then previous studies and limitation then your objective in a precise way.

Rewritten accordingly

3. Ethiopia hosts over 30 distinct indigenous chicken ecotypes and breeds found across different regional districts, including the well-known Horro, Koekoek, Fayoumi, and Tilili populations… What mean indigenous and local chicken ecotypes? Are Fayoumi and Koekoek indigenous to Ethiopia? No, they are local ecotypes. 

Indigenous chicken ecotypes are native breeds that have evolved in specific regions, adapting to local environmental conditions and cultural practices. Local chicken ecotypes encompass both indigenous breeds and those that have been introduced and established in an area. While both types are adapted to local environments and are important for community resources, they differ in origin: indigenous ecotypes are native and locally evolved, whereas local ecotypes may include a mix of indigenous and introduced breeds. Furthermore, indigenous ecotypes typically hold greater cultural significance than local ecotypes, which can vary in their cultural relevance.

Data collection and sample size

1. Thus, a total of 2166 households and 2452 chickens were surveyed in the our kebeles: Corrected in the revised manuscript

2. Please keep the coherence and avoid redundancy. The formula then the samples taken. Corrected in the revised manuscript

Results and Discussion

1. Where are the results of PCA, cluster analysis and body index values in tables/figures

Summary of PCA, Cluster Analysis, and Body Index Values for Indigenous Chickens are well presented in Table 2 of the revised manuscript

General

Still there are problems in editing, coherence, and spelling. Please check throughout the manuscript.

Reviewer 3

The authors of this manuscript "Characterization of Indigenous Chicken Phenotypes in Liban Jawi District, Ethiopia: A Qualitative and Quantitative Analysis". perform a great efforts to try to address both the qualitative and quantitative phenotypes of Liban Jawi district indigenous chicken. This study offers valuable insights into these chicken phenotypes, which will significantly contribute to set an accurate breeding program to improve and enhance these genetic resources in Ethiopia.

However, authors should revise the manuscript one more time to ensure that the grammar, spelling, and overall editing of the manuscript follow the journal's guidelines. We fell that the manuscript is revised to the best of our capacity.

---

## [Decision Letter · Decision Letter 2]

22 Oct 2024

PONE-D-24-28488R2Characterization of Indigenous Chicken Phenotypes in Liban Jawi District, Ethiopia: A Qualitative and Quantitative AnalysisPLOS ONE

Dear Dr. Desalegn Begna

Thank you for submitting your manuscript to PLOS ONE. After careful consideration, we feel that it has merit but does not fully meet PLOS ONE’s publication criteria as it currently stands. Therefore, we invite you to submit a revised version of the manuscript that addresses the points raised during the review process.

Dear Dr., Desalegn Begna,

Thank you for submitting your manuscript to PLOS ONE. After careful consideration, we have decided that your manuscript needs Major Revision.

Kind regards,

Prof. Lamiaa Mostafa Radwan, Ph.D.

Academic Editor

PLOS ONE

**Reviewer 1**

There is a lot of progress compared to the first version, however, much still needs to be improved, especially with discussion

**Reviewer 2**

Comments and Suggestions

Dear authors, I appreciate your efforts to improve the manuscript; however, some issues remain that require attention.

Esteemed authors, I would be pleased if your manuscript were deemed suitable for publication following the rectification of the following remarks and other reviewer’s comments (If any).

1. It would be advantageous for the problem statement, lines 94-96, to precede the objectives section.

2. Please incorporate references pertaining to the description of the study area.

3. Kindly re-evaluate the spelling, grammar, table formatting, and referencing style once more.

4. I remain dissatisfied with Table 2, the PCA, the cluster analysis, and the indices along with the discussion spanning lines 336-342. I would appreciate the inclusion of figures in these multivariate analyses, with morphological indices clearly delineated and analyzed by proposing their functional significance.

5. Please refine the conclusions and recommendations section, as the conclusion appears to function merely as a summary.

**Reviewer 3**

Accept

We look forward to receiving your revised manuscript.

Kind regards,

Lamiaa Mostafa Radwan, Ph.D.

Academic Editor

PLOS ONE

Additional Editor Comments:

Dear Dr., Desalegn Begna,

Thank you for submitting your manuscript to PLOS ONE. After careful consideration, we have decided that your manuscript needs Major Revision.

Kind regards,

Prof. Lamiaa Mostafa Radwan, Ph.D.

Academic Editor

PLOS ONE

Reviewer 1

There is a lot of progress compared to the first version, however, much still needs to be improved, especially with discussion

Reviewer 2

Comments and Suggestions

Dear authors, I appreciate your efforts to improve the manuscript; however, some issues remain that require attention.

Esteemed authors, I would be pleased if your manuscript were deemed suitable for publication following the rectification of the following remarks and other reviewer’s comments (If any).

1. It would be advantageous for the problem statement, lines 94-96, to precede the objectives section.

2. Please incorporate references pertaining to the description of the study area.

3. Kindly re-evaluate the spelling, grammar, table formatting, and referencing style once more.

4. I remain dissatisfied with Table 2, the PCA, the cluster analysis, and the indices along with the discussion spanning lines 336-342. I would appreciate the inclusion of figures in these multivariate analyses, with morphological indices clearly delineated and analyzed by proposing their functional significance.

5. Please refine the conclusions and recommendations section, as the conclusion appears to function merely as a summary.

Reviewer 3

Accept

Reviewers' comments:

Reviewer's Responses to Questions

**Comments to the Author**

1. If the authors have adequately addressed your comments raised in a previous round of review and you feel that this manuscript is now acceptable for publication, you may indicate that here to bypass the “Comments to the Author” section, enter your conflict of interest statement in the “Confidential to Editor” section, and submit your "Accept" recommendation.

Reviewer #1: (No Response)

Reviewer #2: All comments have been addressed

Reviewer #3: All comments have been addressed

2. Is the manuscript technically sound, and do the data support the conclusions?

Reviewer #1: No

Reviewer #2: Yes

Reviewer #3: Yes

3. Has the statistical analysis been performed appropriately and rigorously? 

Reviewer #1: Yes

Reviewer #2: Yes

Reviewer #3: Yes

4. Have the authors made all data underlying the findings in their manuscript fully available?

Reviewer #1: Yes

Reviewer #2: Yes

Reviewer #3: Yes

5. Is the manuscript presented in an intelligible fashion and written in standard English?

Reviewer #1: Yes

Reviewer #2: Yes

Reviewer #3: Yes

6. Review Comments to the Author

Reviewer #1: There is a lot of progress compared to the first version, however, much still needs to be improved, especially with discussion

Reviewer #2: Comments and Suggestions

Dear authors, I appreciate your efforts to improve the manuscript; however, some issues remain that require attention.

Esteemed authors, I would be pleased if your manuscript were deemed suitable for publication following the rectification of the following remarks and other reviewer’s comments (If any).

1. It would be advantageous for the problem statement, lines 94-96, to precede the objectives section.

2. Please incorporate references pertaining to the description of the study area.

3. Kindly re-evaluate the spelling, grammar, table formatting, and referencing style once more.

4. I remain dissatisfied with Table 2, the PCA, the cluster analysis, and the indices along with the discussion spanning lines 336-342. I would appreciate the inclusion of figures in these multivariate analyses, with morphological indices clearly delineated and analyzed by proposing their functional significance.

5. Please refine the conclusions and recommendations section, as the conclusion appears to function merely as a summary.

Reviewer #3: (No Response)

7. PLOS authors have the option to publish the peer review history of their article (what does this mean?). If published, this will include your full peer review and any attached files.

Reviewer #1: No

Reviewer #2: No

Reviewer #3: No

---

## [Author Response · Author response to Decision Letter 2]

26 Oct 2024

Reviewer 1

There is a lot of progress compared to the first version, however, much still needs to be improved, especially with discussion: Revised to the most possible in our capacity

Reviewer 2

Comments and Suggestions

Dear authors, I appreciate your efforts to improve the manuscript; however, some issues remain that require attention.

Esteemed authors, I would be pleased if your manuscript were deemed suitable for publication following the rectification of the following remarks and other reviewer’s comments (If any).

1. It would be advantageous for the problem statement, lines 94-96, to precede the objectives section. We have made this adjustment in the revised version to ensure that the context is clearly established prior to outlining our objectives.

2. Please incorporate references pertaining to the description of the study area. Incorporated 

3. Kindly re-evaluate the spelling, grammar, table formatting, and referencing style once more. Done to our level best

4. I remain dissatisfied with Table 2, the PCA, the cluster analysis, and the indices along with the discussion spanning lines 336-342. I would appreciate the inclusion of figures in these multivariate analyses, with morphological indices clearly delineated and analyzed by proposing their functional significance. Done accordingly in the revised document

5. Please refine the conclusions and recommendations section, as the conclusion appears to function merely as a summary. conclusions and recommendations section revised to the bests of our capacity

Reviewer 3

Accept

---

## [Decision Letter · Decision Letter 3]

4 Nov 2024

Characterization of Indigenous Chicken Phenotypes in Liban Jawi District, Ethiopia: A Qualitative and Quantitative Analysis

PONE-D-24-28488R3

Dear Dr. Begna,

We’re pleased to inform you that your manuscript has been judged scientifically suitable for publication and will be formally accepted for publication once it meets all outstanding technical requirements.

Kind regards,

Lamiaa Mostafa Radwan, Ph.D.

Academic Editor

PLOS ONE

Additional Editor Comments (optional):

Accept

Reviewers' comments:

Reviewer's Responses to Questions

**Comments to the Author**

1. If the authors have adequately addressed your comments raised in a previous round of review and you feel that this manuscript is now acceptable for publication, you may indicate that here to bypass the “Comments to the Author” section, enter your conflict of interest statement in the “Confidential to Editor” section, and submit your "Accept" recommendation.

Reviewer #2: All comments have been addressed

Reviewer #3: All comments have been addressed

2. Is the manuscript technically sound, and do the data support the conclusions?

Reviewer #2: Yes

Reviewer #3: Yes

3. Has the statistical analysis been performed appropriately and rigorously? 

Reviewer #2: Yes

Reviewer #3: Yes

4. Have the authors made all data underlying the findings in their manuscript fully available?

Reviewer #2: Yes

Reviewer #3: Yes

5. Is the manuscript presented in an intelligible fashion and written in standard English?

Reviewer #2: Yes

Reviewer #3: Yes

6. Review Comments to the Author

Reviewer #2: Dear Authors, Thank you for the improvement of your manuscript. Please delete the statement ''Therefore, this study aims to phenotypically characterize

83 indigenous chickens in the Liben Jawi district, focusing on both qualitative and

84 quantitative traits, to document their unique physical characteristics and identify

85 desirable traits for genetic improvement, thus laying the groundwork for effective

86 conservation and sustainable utilization of these valuable local poultry genetic

87 resources''. Which is redundant.

Reviewer #3: I think the authors carefully addressed all reviewer comments.

The manuscript significantly improved.

7. PLOS authors have the option to publish the peer review history of their article (what does this mean?). If published, this will include your full peer review and any attached files.

Reviewer #2: No

Reviewer #3: No

---

## [Editor Report · Acceptance letter]

12 Nov 2024

PONE-D-24-28488R3 

PLOS ONE

Dear Dr. Begna, 

I'm pleased to inform you that your manuscript has been deemed suitable for publication in PLOS ONE. Congratulations! Your manuscript is now being handed over to our production team.

Kind regards, 

on behalf of

Prof. Dr. Lamiaa Mostafa Radwan 

Academic Editor

PLOS ONE